

# Sensitivity of mid-Pliocene climate to changes in orbital forcing, and PlioMIP's boundary conditions

Eric Samakinwa[1,2,3], Christian Stepanek[1], and Gerrit Lohmann[1]

[1]Alfred Wegener Institute (AWI), Helmholtz Centre for Polar and Marine Research, Bremerhaven, Germany.
[2]Institute of Geography, University of Bern, Switzerland.
[3]Oeschger Centre for Climate Change Research, University of Bern, Switzerland.

**Correspondence:** Eric Samakinwa (eric.samakinwa@giub.unibe.ch)

**Abstract.** In this study, we compare results obtained from modelling the mid-Pliocene warm period using the Community Earth System Models (COSMOS, version: COSMOS-landveg r2413, 2009) with the two different modelling methodologies and sets of boundary conditions prescribed for the two phases of the Pliocene Model Intercomparison Project (PlioMIP), tagged PlioMIP1 and PlioMIP2. Boundary conditions, model forcing, and modelling methodology for the two phases of PlioMIP differ

considerably in palaeogeography, in particular with regards to the state of ocean gateways, ice-masks, treatment of vegetation and topography. Further differences between model setups as suggested for PlioMIP1 and PlioMIP2 consider updates to the concentration of trace gases: atmospheric carbon dioxide ($CO_2$), is specified as 405 and 400 parts per million by volume (ppmv) for PlioMIP1 and PlioMIP2, respectively. There are also minor differences in the concentrations of methane ($CH_4$) and nitrous oxide ($N_2O$) due to changes in the protocol of the Paleoclimate Model Intercomparison Project (PMIP) from phase 3 to

phase 4. Employing a single model across two phases of PlioMIP enables a better understanding of the impact that the various differences in modelling methodology between PlioMIP1 and PlioMIP2 have on model output. Yet, a dedicated comparison of COSMOS model output of PlioMIP1 and PlioMIP2 is not in the curriculum of model analyses proposed in PlioMIP2. Here, we bridge the gap between our contributions to PlioMIP1 (Stepanek and Lohmann, 2012) and PlioMIP2 (Stepanek et al., 2020). We highlight some of the effects that differences in the chosen mid-Pliocene model setup (PlioMIP2 vs. PlioMIP1) have on

the climate state as derived with the COSMOS, as this information will be valuable in the framework of the model-model and model-data-comparison within PlioMIP2. We evaluate the model sensitivity to improved mid-Pliocene boundary conditions using PlioMIP's core mid-Pliocene experiments for PlioMIP1 and PlioMIP2, and present further simulations where we test model sensitivity to variations in palaeogeography, orbit and concentration of $CO_2$.

Firstly, we highlight major changes in boundary conditions from PlioMIP1 to PlioMIP2 and also the challenges recorded from

the initial effort. The results derived from our simulations show that COSMOS simulates a mid-Pliocene climate state that is 0.29 K colder in PlioMIP2, if compared to PlioMIP1 (17.82 °C in PlioMIP1, 17.53 °C in PlioMIP2, values based on simulated surface skin temperature). On one hand, high-latitude warming, which is supported by proxy evidence of the mid-Pliocene, is underestimated in simulations of both PlioMIP1 and PlioMIP2. On the other hand, spatial variations in surface air temperature (SAT), sea surface temperature (SST) as well as the distribution of sea ice suggest improvement of simulated SAT and SST

in PlioMIP2 if employing the updated palaeogeography. Our PlioMIP2 mid-Pliocene simulation produces warmer SSTs in





the Arctic and North Atlantic Ocean than derived from the respective PlioMIP1 climate state. The difference in prescribed $CO_2$ accounts for 1.1 K of warming in the Arctic, leading to an ice-free summer in the PlioMIP1 simulation, and a quasi ice-free summer in PlioMIP2. Beyond the official set of PlioMIP2 simulations, we present further simulations and analyses that sample the phase space of potential alternative orbital forcings that have acted during the Pliocene and may have impacted

on geological records. Employing orbital forcing, which differ from that proposed for PlioMIP2 (i.e. corresponding to Pre-Industrial conditions) but falls into the Mid-Pliocene time period targeted in the PlioMIP, leads to pronounced annual and seasonal temperature variations, which are not directly retrievable from the marine and terrestrial reconstruction of the time-slice.

## 1 Introduction

In 2050, global population is expected to have increased by 2.7 billion relative to its 2005 value (Bongaarts, 2009). In conjunction with this population increase, energy demand is also rising. While short-term perspectives for a carbon-neutral global economy and effective $CO_2$ draw-down technologies are absent, a direct implication of population dynamics for the climate system is increased concentration of atmospheric $CO_2$ through anthropogenic activities, increased waste heat from energy production that hampers the local climate of some parts of the world, loss of polar sea ice and last but not the least, rising

global temperature. Through the continuous release of trace gases that impact on the Earth's energy balance such as $CO_2$ into the atmosphere, and as a result of the vast thermal inertia of the Earth system, future warming is inevitable. In expectation of a warmer climate, it is of utmost importance to quantify climatic conditions that humankind could face in the near future, in order to enable proper preparation and to inform about the need and possibility of mitigation measures, wherever possible. The mid-Pliocene Warm Period (mPWP) (3.264 - 3.025 Million years (Ma) Before Present (BP); Dowsett et al. (2016)) has been

suggested as a time-slice which could provide a possible insight to future climate in terms of temperature (Jansen et al., 2007). Climate model simulations of this time-slice suggest global mean temperatures which are 2 - 3 °C higher than today (Haywood and Valdes, 2004; Jansen et al., 2007) with land surface conditions, continental configuration, and atmospheric concentrations of $CO_2$ that were similar, albeit not identical to present-day (Raymo et al., 1996; Pagani et al., 2010; Kürschner et al., 1996; Haywood et al., 2016). Evidence from the geologic record suggests that in the past the Arctic was more vegetated than today,

e.g. during the Mid-Pliocene warm period (Rybczynski et al., 2013) and the Pleistocene-Pliocene (Matthews and Fyles, 2000). These records of the past offer us a glimpse into a climate of increased temperatures in polar regions that may return in the next decades (Overland et al., 2014) as an effect of human influence on climate. Hence, the mid-Pliocene can be considered a useful, but not direct analogue for future warmth (Jansen et al., 2007).

Characteristics of mid-Pliocene climate can be inferred either through records of past climate that were stored in geological

archives, or by exposing climate models to boundary conditions and model forcing representative of the Earth system characteristics of that time-slice. These may include altered continental configuration, past land elevation and ocean bathymetry as well as the atmospheric composition and orbital configuration representative for the time-slice under study. Furthermore, a parameterization of altered vegetation distribution may be necessary if such is not dynamically computed by the model itself.



Creating boundary conditions for past time periods is one of the most time consuming tasks faced in palaeoclimate modelling. Assumptions with respect to implementation of details of palaeoclimate boundary conditions can vary amongst researchers, and the mid-Pliocene is not an exceptional time-slice in this respect. To provide common grounds for the model intercomparison in simulating the mPWP, and to reduce the need for repeating simulations with updated model setups, the Pliocene

Modelling Intercomparison Project (PlioMIP) provides for phase 1 (PlioMIP1, Haywood et al., 2010, 2011) and 2 (PlioMIP2, Haywood et al., 2016) sets of boundary conditions that are implemented consistently across the model ensemble.

As shown in the framework of PlioMIP1 (e.g. Haywood et al., 2013), the PlioMIP provides a unique opportunity to reconcile our knowledge about the mechanisms in, and the characteristics of, a warmer climate. In this framework, consistency of various climate models' simulations of the mPWP and their ability in reproducing climate patterns that have been inferred from data

stored in geological climate archives is sampled and compared in a coordinated effort. To support this ambition in PlioMIP2, we provide in this manuscript important information that relates Community Earth System Models (COSMOS) simulations, contributed to both PlioMIP1 (Stepanek and Lohmann, 2012) and PlioMIP2 (Stepanek et al., 2020), to each other. In this effort we study how differences in the boundary conditions and model forcing, that are present between COSMOS simulations for PlioMIP1 and PlioMIP2 as a result of updates to the relevant PlioMIP/PMIP protocols, influence various large scale climate

patterns. To provide boundary conditions for the climate models, and to derive model-independent paleoclimate information, that can be employed for a model-data-comparison, PlioMIP works in alliance with the US Geological Survey's Pliocene Research Interpretation and Synoptic Mapping (PRISM) project that has constantly improved the data basis for Pliocene paleoenvironment over the last 25 years (e.g. Dowsett et al., 2013). All model data presented in this manuscript is hence directly based on output from the PRISM project. Following the conclusion of PlioMIP1, PlioMIP2 utilizes state-of-the-art boundary

conditions that have emerged over the last few years, with updated reconstructions of ocean bathymetry and land-ice, surface topography, as well as new datasets describing the distribution of mid-Pliocene soils and lakes (Haywood et al., 2016). With the exception of the lake reconstruction, all these datasets are employed in our PlioMIP2 simulations (Stepanek et al., 2020). Lakes cannot be adequately represented in the employed model setup of the COSMOS (Stepanek et al., 2020). While this reduces comparability between COSMOS and other models in PlioMIP2 that employ the full set of boundary conditions pro-

vided in the enhanced dataset (Haywood et al., 2016), it improves comparability between COSMOS simulations of PlioMIP2 and PlioMIP1, where also in the latter lakes are not considered. Here, we specifically address improved paleogeography and the change of mPWP concentrations of atmospheric $CO_2$ from PlioMIP1 to PlioMIP2. Furthermore, we take the opportunity to go beyond the PlioMIP2 protocol (Haywood et al., 2016) and sample the impact of alternative orbital forcing, that could have shaped the reconstructed climate over the time period from 3.26 to 3.025 Ma BP, on our modelled climate state. Our aim

is to test the extent to which alternative configurations of the Earth's orbit during the mPWP could potentially improve the agreement of modelled and reconstructed mPWP climate states.





## 2 Methodology

### 2.1 Evolution of mid-Pliocene model Palaeogeography and forcing

Palaeogeographic boundary conditions for simulating the Pliocene have undergone major changes from PlioMIP1 to PlioMIP2. The switch from PRISM3D (Dowsett et al., 2010) to PRISM4 (Dowsett et al., 2016) has led to the introduction of various

changes in the mPWP model setup of the COSMOS. Updates to topography and bathymetry reflect changes in dynamic topography, global isostatic adjustment, as well as new findings that suggest a reduced extent of the Greenland Ice Sheet (Haywood et al., 2016; Dowsett et al., 2016). The PRISM4 Antarctic ice-sheet on the other hand remains over East Antarctica consistent with the version suggested by PRISM3D (Haywood et al., 2016, and references therein) (compare Figure 1a and b). A major difference between the PlioMIP2 and PlioMIP1 model setup is the configuration of two ocean gateways – the Bering

Strait, that is closed in PlioMIP2 by separating Bering Sea and Chukchi Sea with a land bridge, and the Canadian Arctic Archipelago – both were open in PlioMIP1 (compare Figure 1a and b). Based on previous work, closing these gateways increases modelled Sea Surface Temperature (SST) in the North Atlantic Ocean, which would reduce the apparent disagreement of models and reconstructions in these region (Dowsett et al., 2013; Otto-Bliesner et al., 2016). Closing the Bering Strait is supported by evidence that this gateway may have been temporarily closed in proximity to the mPWP (Matthiessen et al.,

15 2008).

High concentration of atmospheric $CO_2$ is often presented as one of the major reasons for the relatively higher than present-day temperatures during the mid-Pliocene (Seki et al., 2010; Pagani et al., 2010). However, the exact concentration of atmospheric $CO_2$ during this time-slice remains uncertain with various values being suggested (Raymo et al., 1996; Kürschner et al., 1996; Haywood et al., 2010, 2011, 2016), which complicates a comparison between models and reconstructions. Within PlioMIP,

the absence of reconstructions for other radiative-active trace gases is acknowledged, and the respective radiative forcing is absorbed into an increased value of $CO_2$ – yet, this value differs between PlioMIP1 and PlioMIP2 (Haywood et al., 2010, 2016). Furthermore, between PlioMIP1 and PlioMIP2 the concentrations of the Pre-Industrial (PI) control state of the COSMOS changed due to an update of the reference setup to the PMIP4 protocol (Otto-Bliesner et al., 2017), implying small differences in the employed concentrations of methan and nitrous oxide between PlioMIP2 and PlioMIP1. While these changes are small,

we do not want to neglect their potential effect on climatic differences between PlioMIP2 and PlioMIP1 COSMOS simulations. Therefore, here we study the effect of differences between the trace-gas forcing of both iterations of PlioMIP on the climate of the mPWP as simulated with COSMOS.

One additional topic addressed in this study is the potential for varying orbital forcing that may have influenced the mPWP over time. Both PlioMIP1 and PlioMIP2 assume a modern orbit for simulations of the mPWP (Haywood et al., 2010, 2011, 2016).

This choice represents only one of multiple orbits that were present during the PlioMIP period, spanning Marine Isotope Stage (MIS) G21 to M1 (Haywood et al., 2016). Prescribing present-day orbital forcing, that is very close to the configuration during the KM5c time slice selected for PlioMIP2 (Haywood et al., 2016), may not reproduce the expected warm/cold condition that could have been enabled by an orbit that occurred sometime during the mPWP. This is a side effect of computing an average mPWP condition rather than attempting to study the transient climate evolution of the mPWP within PlioMIP – such





study would be a promising approach to identify orbital-forced warm peaks, but is beyond the capabilities of the model-intercomparison. Most of the mis-matches recorded between mid-Pliocene model simulations and interpretation of data stored in geological archives have been attributed to the choice of orbital forcing prescribed for model simulations Haywood et al. (2013b); Prescott et al. (2014). The need for a more discrete time slice for simulating the mPWP towards an improved

model-data comparison has been stated (Haywood et al., 2013). Thus, the KM5c time-slice has been selected, partly on the basis of a strong similarity of the orbit at that time to the modern orbital configuration, that is useful towards interpreting paleoenvironments in the context of future warming (Haywood et al., 2016) that is based on antropogenic activity and will obviously be set in a near-modern orbital configuration. While acknowledging the utility of the KM5c orbit for the scientific aims of PlioMIP2, here we go beyond the simulation of the KM5c and quantify the effect of an alternative orbital configuration.

We create model setups where the prescribed PlioMIP2 model setup for the COSMOS employs an orbital configuration that is representative of MIS K1. The MIS K1 time-slice is one of the lightest isotope excursions found during the mPWP with the total global annual mean insolation being is approximately 0.5 W/m$^2$ higher than modern (Prescott et al., 2014). The effect of orbital forcing on the mPWP has been studied by Prescott et al. (2014), with emphasis on Surface Air Temperature (SAT). The novelty in our approach is that we employ the updated PlioMIP2 model setup and test the impact of orbital configuration

also on the ocean state, in particular on SST and sea ice. Furthermore, we put differences in orbital forcing into context with varying concentrations of $CO_2$.

## 2.2 Model description COSMOS

The coupled atmosphere-ocean model used in producing simulations for this study is the COSMOS, which was developed by the Max Planck Institute for Meteorology (MPI) in Hamburg, Germany. The COSMOS consists of four major components,

namely the ECHAM5 atmosphere model (Roeckner et al., 2003), the MPI-OM ocean model (Marsland et al., 2003), and the land-vegetation and carbon cycle model JSBACH (Raddatz et al., 2007). For a description of the coupled setup and an evaluation of its performance please refer to Jungclaus et al. (2006). The ocean biogeochemistry model HAMOCC, introduced by Maier-Reimer (1993), is as well a part of the COSMOS, but is not used in the production of PlioMIP simulations. The COSMOS has already proven to be a valuable tool for the study of paleoclimate, also beyond the Pliocene epoch. The various

time-slices studied by means of the COSMOS include, but are not limited to, the Last Millennium (Jungclaus et al., 2010), warm climates of the Miocene (e.g., Knorr et al., 2011), the mid-Pliocene (Stepanek and Lohmann, 2012), as well as glacial (e.g., Gong et al., 2013; Kageyema et al., 2013; Zhang et al., 2013) and interglacial climates (e.g., Pfeiffer et al., 2016; Varma et al., 2012; Wei and Lohmann, 2012). A detailed description of the COSMOS model components is given by Stepanek and Lohmann (2012).

## 2.3 Experimental designs

The aim of this study is to identify and discuss differences in the modelled climate that occur between the COSMOS simulations contributed to PlioMIP1 and PlioMIP2, and to study the effect of orbits warmer than the configuration applied for PlioMIP. With this aim in mind, we have applied a methodology for setting up simulations as described below. In general, experiments





were carried out following PlioMIP1 and PlioMIP2 protocols (Haywood et al., 2010, 2011, 2016), respectively. Yet, for the purpose of this study, small modifications to the proposed official PlioMIP model setups are necessary. Three different orbits are employed here (Table 1). Two of them are similar and representative of modern, respectively Pre-Industrial, conditions – the only difference between them being small deviations that were introduced into the reference setup of the COSMOS between

PlioMIP1 and PlioMIP2 via adaptation of the Pre-Industrial orbit in COSMOS according to the PMIP4 protocol (Otto-Bliesner et al., 2017). The third employed configuration represents the orbit of MIS K1, the values of orbital parameters being based on the astronomical solution by Laskar et al. (2004). Earth's orbital parameters are prescribed as constant values of eccentricity, obliquity and the longitude of perihelion as outlined in Table 1. The orbit employed for simulating K1 is consistent with the configuration chosen by Prescott et al. (2014).

Simulations are classified in four different categories namely, Pre-Industrial, standard PlioMIP1 set-up, standard PlioMIP2 set-up, and modified PlioMIP2 set-up with adapted (MIS K1) orbit (PlioMIP2_K1). First in the order as shown in Table 1 is the PI category, which consists of the PI-control simulations (PI_1 and PI_2) for PlioMIP1 and PlioMIP2, respectively. Ocean bathymetry and land-sea mask are taken from the standard modern setup of the COSMOS. As described by Jungclaus et al. (2006), this set-up has been generated based on the Earth Topography Five Minute Grid (ETOPO5, National Geophysical

Data Center, 1988; Edwards et al., 1992). An identical $CO_2$ concentration of 280 ppmv is prescribed for PI_1 and PI_2, but both experiments differ slightly with respect to orbital parameters and volume mixing ratios of trace gases $N_2O$ and $CH_4$ (see Table 1). The constant volume mixing ratios of 270 ppbv $N_2O$ and 760 ppbv $CH_4$ are prescribed for PI_1, while the corresponding values for PI_2 are set to 273 and 808 ppbv, respectively. Chlorofluorocarbons on the other hand are constant across all simulations and are absent. Category PlioMIP1 consists of simulation PlioM1, which is the COSMOS mid-Pliocene

experiment within the framework of PlioMIP1 (Stepanek and Lohmann, 2012) and utilizes the PRISM3D land-sea mask, orography and ice-mask. Category PlioMIP2 consists of mid-Pliocene simulations based on PRISM4 boundary conditions with slight differences in either orbital forcing or concentration of atmospheric $CO_2$. Simulation Eoi400 is the COSMOS mid-Pliocene experiment for PlioMIP2 (Stepanek et al., 2020) in which $CO_2$ is prescribed to be 400 ppmv, while simulation Eoi405 is derived from Eoi400 in that the concentration of carbon dioxide is set to the PlioMIP1 $CO_2$ forcing of 405 ppmv.

In comparing them, both simulations enable us to study the impact of the difference of $CO_2$ forcing between the two phases of PlioMIP on achieved results. Furthermore, simulation Eoi400_ORB employs the orbital forcing utilized by COSMOS in simulating the mid-Pliocene for PlioMIP1, while retaining other boundary conditions and forcing as prescribed for PlioMIP2. Direct comparison between simulations Eoi400 and Eoi400_ORB will give an indication of the influence of the slight change in orbital forcing on mid Pliocene climate as simulated with COSMOS for PlioMIP1 and PlioMIP2. This is important within the

framework of PlioMIP, where orbital forcing is described as similar to modern day, but specific Pre-Industrial orbital parameters differ in the case of the COSMOS between PlioMIP1 and PlioMIP2. Simulation Eoi405_ORB of category PlioMIP2 differs from simulation PlioM1 (Stepanek and Lohmann, 2012) in that it employs the paleoenvironmental reconstruction of PRISM4, and also employs different trace house gas concentrations for methane and nitrous oxide. It differs from simulation Eoi400 on the other hand in both orbital configuration and the prescribed concentration of carbon dioxide (see Table 1). This choice of

modelling methodology enables us to infer the relative effects of the improved representation of mid-Pliocene geography from





PlioMIP1 to PlioMIP2 on our model, while honouring the presence of other differences in the model setup. Furthermore, in order to study the effect of an alternative orbit on mid-Pliocene warmth, simulation Eoi400_K1 in category PlioMIP2_K1 is consistent with the standard mid-Pliocene set-up prescribed for PlioMIP2 and employed in simulation Eoi400, but employs a different orbital forcing which is representative of MIS K1. The choice of our experimental design, enabling a direct comparison

between simulations Eoi400 and Eoi400_K1, provides an indication of orbitally influenced climatic variability within the mid-Pliocene. Ultimately, the total effect of all the improved boundary conditions in PlioMIP2 is examined by comparing simulations PlioM1 and Eoi400.

## 3 Results

This section present results of the mid-Pliocene and PI simulations listed in Table 1, and attempt to investigate and quantify the

differences in the mid-Pliocene climate simulations by COSMOS for PlioMIP1 and PlioMIP2, respectively. Furthermore, we present results from our sensitivity experiments which show the relative contributions of newly prescribed PlioMIP2 boundary conditions (Haywood et al., 2016), with respect to that of PlioMIP1 (Haywood et al., 2010, 2011) in the context of deviations of the COSMOS model setup between PlioMIP1 and PlioMIP2 that are not due to the PlioMIP2 protocol. We also show simulated seasonal variability which may occur due to orbital forcing prescribed in simulating the mid-Pliocene climate. This

is examined by a direct comparison of climate forced by two distinct orbital configurations representative of two discrete time-slices within the mid-Pliocene, namely MIS K1 and the PlioMIP2 reference orbit of MIS KM5c. The MIS KM5c is selected for simulations within the framework of PlioMIP2 due to it's strong orbital similarity to present-day (Haywood et al., 2016) and thus present-day orbital configuration is adopted (Otto-Bliesner et al., 2017). We analyse and report in this section annual and seasonal means of SAT, SST, and sea ice distribution. Anomalies are tested with regard to significance in the context of

internal variability in the contributing simulations by means of the autocorrelation method by Matalas and Dawdy (1964).

### 3.1 SAT

The physical quantity SAT is defined in this study as the temperature of air near the surface (below 2 meter) of the Earth. At large scale, the COSMOS simulate fairly similar patterns of mid-Pliocene SAT in response to boundary conditions for PlioMIP1 and PlioMIP2 core experiments. Yet, there are noteworthy differences in details of mid-Pliocene SAT anomalies

between PlioMIP2 and PlioMIP1. The most pronounced annual average SAT anomaly of mPWP relative to PI occurs in both phases of PlioMIP at high latitudes and in polar regions, providing evidence of a similar, albeit not identical, level of polar amplification in PlioMIP1 and PlioMIP2 mPWP model setups. On average, the COSMOS simulate a mid-Pliocene climate that is 0.29 K colder in PlioMIP2 with the global average SAT reaching 290.97 K for PlioMIP1, while the corresponding value for PlioMIP2 is 290.68 K. It is important to note that the values computed for PlioMIP1 are recomputed based on averaging

over 100 years, while Stepanek and Lohmann (2012) have, in agreement with the PlioMIP1 analysis protocol, provided averages over 30 years. Anomalies between coupled ocean-atmosphere simulations averaged over 100-year period for PlioMIP1 and PlioMIP2, namely PlioM1 and Eoi400, show that SAT anomalies are fairly similar in PlioMIP1 and PlioMIP2 over the



equatorial oceans, but that there is substantial deviation of SAT over continents and polar regions if results from PlioMIP1 and PlioMIP2 are compared with each other. The most pronounced warming from PlioMIP1 to PlioMIP2 is seen over Greenland and Antarctica. On the other hand, gradual cooling is present over the Arctic from PlioMIP1 to PlioMIP2. Over the oceans in PlioMIP2, COSMOS simulates warmer SAT over the North Atlantic, while SAT over the Indian Ocean, South Atlantic Ocean,

and low-latitude Pacific Ocean is largely unchanged. Generally, both PlioMIP1 and PlioMIP2 mid-Pliocene simulations suggest that land masses are warmer than the ocean, and furthermore that there is presence of substantial polar amplification; the latter is more pronounced in PlioMIP1 simulation PlioM1 (compare Figure 2a, 2b and 3b). The strong regional warming simulated over Greenland and Antarctica is linked to changes in albedo and orography over these regions (confer Figure 1a and 1b). As the magnitude of changes in albedo and elevation over Greenland are more pronounced in PlioMIP2 than in PlioMIP1 (Figure

1a,b) due to reduction in the spatial extent of Greenland ice-sheet from PRISM3D to PRISM4 (Dowsett et al., 2016), also the regional warming signal is more pronounced in PlioMIP2 (Figure 3b). For simulation PlioM1, a warming of about 15 K is found over Greenland, while the corresponding value is about 21 K for Eoi400. This illustrates that the mPWP SAT anomaly simulated with the COSMOS is regionally increased in PlioMIP2 in comparison to PlioMIP1, even though the global average mPWP temperature anomaly is smaller in PlioMIP2. Antarctic ice-sheet estimates, based on results produced with the British

Antarctic Survey Ice Sheet Model utilizing a climate simulation produced with PRISM boundary conditions (Haywood et al., 2010), remain the same for both phases of PlioMIP (Dowsett et al., 2016; Haywood et al., 2016). Over Antarctica, there are strong regional mPWP temperature anomalies, while cooling is evident in the South Pacific, between 60°S - 70°S, extending from 65°W - 150°W for both simulations. This South Pacific cooling is about -1.2 K on average in PlioMIP1 (PlioM1 with respect to PI_1), but more intense in PlioMIP2 (Eoi400 vs. PI_2), where the temperature anomaly locally reaches -4 K and is

characterized by a wider spatial extent. In both PlioMIP1 and PlioMIP2, the Indian sector of the Southern Ocean experiences intense warming that extends from 60°E into the South Pacific Ocean. Extreme values of SAT anomalies vary from 25.2 K in PlioM1 to 23.1 K in Eoi400 over the adjacent Antarctic landmass. Moving the focus again to the Northern Hemisphere, we highlight that both simulations PlioM1 and Eoi400 are characterized by land-sea masks in which the modern Hudson Bay is absent, while that region is part of the oceans in our PI simulation setups. The change in the land sea mask introduces

an annual mean warming in the mid-Pliocene in comparison to Pre-Industrial that is fairly similar in PlioMIP1 (simulation PlioM1) and PlioMIP2 (simulation Eoi400), but slightly stronger in PlioMIP1.If we study the influence of modified radiative forcing on mPWP SAT, that is introduced by differences in prescribed $CO_2$ between PlioMIP1 and PlioMIP2, and we find a contrasting effect over the Arctic and Antarctica Figure 4a. The prescribed 5 ppmv difference between simulations Eoi405 and Eoi400 causes cooling of the Arctic region and a dipole of warming and cooling over the North Atlantic, while the rest of

the ocean surface warms up. Testing the impact of the small deviation in orbital forcing on the mPWP climate we find that, despite prescribing orbital forcing that is representative of modern day for both PlioMIP1 and PlioMIP2, small changes in this forcing reflect different patterns of warming in the polar regions, as well as in mid-to-high latitudes. The PlioMIP2 core experiment Eoi400, if forced in simulation Eoi400_ORB with orbital configuration utilized by (Stepanek and Lohmann, 2012) in PlioMIP1, shows fairly constant SAT in the Equatorial Pacific, while it is relatively warmer in the Atlantic and Indian Ocean

(see Figure 4b).



Analyzing the effect of a different orbital configuration that is known to have been present during the mPWP we find via a comparison between simulations Eoi400_K1 and Eoi400 that specifying orbital parameters, representative of MIS K1, will produce a mid-Pliocene climate that is warmer at low-and-high latitudes and in the Arctic, but that is colder in the Southern Hemisphere polar region (see Figure 5a). While almost all land masses are warming under the influence of MIS K1 orbital

forcing, cooling is evident over the southern part of the modern United States, north Greenland, northern Australia and some parts of Eurasia. With the exception of the impact of eccentricity, orbital forcing causes a redistribution of incoming solar radiation across latitudes and seasons, rather than a change of the overall input of solar radiation into the climate system. Hence it is of particular interest to study the seasonality of the climatic effect. We find that the impact of MIS K1 orbital forcing is stronger at seasonal time scale than in the annual mean. We show time-mean SAT anomalies for boreal summer (JJA) and

boreal winter (DJF) in order to analyse the orbital control on seasonality of the mid-Pliocene (Figure 5b and c). Our finding is that in simulating the mid-Pliocene, the choice of orbit has great influence on seasonal patterns of SAT. During boreal winter, the K1 orbit is warmer with respect to KM5c, with a maximum SAT anomaly of 3.5 K in comparison to the simulation with KM5c orbit, while averages over boreal summer months provide a colder Northern Hemisphere for Eoi400_K1. Substantial winter warming is present over Eurasia, North and South America as well as South-West Africa. SAT over Greenland shows

pronounced seasonal variation, with warming during winter and cooling in summer. We note potential implications of relative summer cooling for the state of the Greenland Ice Sheet during the mPWP.

With respect to the impact of K1 orbital forcing on mPWP climate in the Southern Hemisphere we find that the seasonal dependency of the temperature anomaly is different for Antarctica, where cooling is dominant all year round. SAT changes in the Southern Hemisphere during summer are not statistically significant in many regions, based on the significance test to

account for effective degrees of freedom by means of the autocorrelation method of Matalas and Dawdy (1964). This indicates similarity between summer SAT simulated in this region for both K1 and KM5c (see Figure 5b). We suggest that changes in palaeogeography between the two phases of PlioMIP may be the main contributor to the differences between mid-Pliocene SAT for PlioMIP1 and PlioMIP2, as simulated by the COSMOS. Simulations conducted to account for the influence of the PRISM4 geography show interestingly a similar pattern with the total boundary condition effect, the latter deduced from the

difference between simulations Eoi400 and PlioM1 (compare Figure 3a and 3b). The major difference between the effect of changed palaeogeography and the total effect (including orbit) is noticed in the Arctic region, where cooling of a relatively higher magnitude is observed for the total effect. Furthermore, there are significant differences between these simulations especially in regions where obvious changes in orography are applied.

## 3.2  SST

In mid-Pliocene simulations for both PlioMIP1 and PlioMIP2, the equatorial warm pool extends across all ocean basins of the world (Figure 6). This warm pool is characterized as a pattern of warm water at the surface around the equator, and defined as the region where absolute SSTs exceed 28.5°C (Watanabe, 2008). In COSMOS simulations we find no change in the pattern of the equatorial warm pool between the mid-Pliocene simulations of PlioMIP2 and PlioMIP1, as both realizations of the



mid-Pliocene climate state show warm pools of almost identical spatial extent (cf. Figure 6). Beyond the extent of the equatorial warm pool we find similarity of SST anomalies obtained from the comparison of mid-Pliocene simulations of PlioMIP1 (PlioM1) and PlioMIP2 (Eoi400) to their respective PI control runs, that show in many regions of the world very similar cooling and warming signals. The Northern Hemisphere surface ocean is – with few exceptions of regional cooling – characterized by

regions of increased SST, a pattern that is more pronounced in the Pacific. Evidence from PlioMIP1 (Dowsett et al., 2013) suggests that mid-Pliocene North-Atlantic SST, as derived from geological records, is consistently underestimated by the models. The COSMOS PlioMIP2 mid-Pliocene simulation produces a warmer North-Atlantic than the respective PlioMIP1 simulation (compare Figure 6 a,b). As a result, the model-data-discord in that region, that has been identified by Dowsett et al. (2013), is confirmed by obtaining the Root Mean Square Deviation (RMSD) between COSMOS

simulated SSTs for both PlioMIP1 and PlioMIP2 and the SST reconstruction by Dowsett et al. (2013) values taken from their supplementary table S1), respectively. A RMSD of 5.14 is evident between reconstructed and modelled mid-Pliocene North Atlantic SSTs in PlioMIP1 (Stepanek and Lohmann, 2012), while a corresponding RMSD of 2.96 is estimated in PlioMIP2 (Stepanek et al., 2020). A better agreement between reconstructed and simulated North Atlantic SSTs for PlioMIP2 is linked to the strength of the Atlantic Meridional Overturning Circulation (AMOC) at different depths. Increased SSTs are accompanied

by enhanced AMOC in the upper cell at about 1,000 m depth, which transports more heat to high latitudes of the Northern Hemisphere (cf. Figure 7 a,b). Furthermore, the inflow of deep-water from the South-Atlantic, more precisely the transport of the Antarctic Bottom Water, is stronger in our PlioMIP2 simulation: AMOC at depths shallower than 3000 m is slightly enhanced from PlioMIP1 to PlioMIP2 (see Figure 7a and b). Similarly, RMSD difference of 0.51 is obtained in favor of Eoi400 when compared with Eoi400_K1, observing a clearer signature of MIS KM5c than K1 in the available mid-Pliocene SST

reconstruction.

Similar to SAT, COSMOS simulates almost the same SST pattern in PlioMIP1 and PlioMIP2 when comparing the influence of PRISM4 geography to the total boundary condition effect (constituting geography, orbit, and atmospheric trace gases). Both mid-Pliocene simulations show the expected warming in the Arctic Ocean, North Atlantic Ocean, and also in the Indian and Atlantic sector of the Southern Ocean. In contrast to the total effect of PlioMIP2 boundary conditions, PRISM4 geography pro-

duces a warmer Equatorial Pacific when implemented instead of PRISM3D geography together with other PlioMIP1 boundary and initial conditions (see Figure 8a and b). The relative contribution of $CO_2$ difference between PlioMIP1 and PlioMIP2 generally produces a warmer ocean surface in PlioMIP1 in comparison to the setup of PlioMIP2, which is more pronounced in the Southern Ocean (see Figure 9a). In the North Atlantic, a dipole of warm and cold ocean surface prevails as seen in SAT, with the maximum SST increase amounting to 2.9°C, while maximum cooling reaches -1.3°C as a result of increased $CO_2$.

The North Atlantic pattern is largely similar when considering the effect of changes in orbital forcing on mid-Pliocene SSTs (Figure 9b), with the obvious difference being a modification of the warming/cooling pattern and decreased magnitude of SST anomalies. If we turn our attention towards the influence of K1 orbital forcing on simulated SSTs of the mPWP, we find that a maximum anomaly of 2.2°C is obtained from the difference between simulations Eoi400_K1 and Eoi400 in the mid-latitudes (Figure 10). Furthermore, simulation Eoi400_K1 with K1 orbital forcing produces warmer Northern Hemisphere oceans than

that of simulation Eoi400 that is based on a KM5c orbit – with the few but obvious exceptions being the Arctic Ocean beyond





the Barents Sea, Greenland Sea, Hudson Bay, as well as the central North Pacific Ocean, where (regional) cooling is observed. The Southern Ocean on the other hand shows mixed signals of warming and cooling under the influence of K1 orbital forcing, with the most pronounced SST increase noticed to the south east of Australia and South America, which is probably related to a shift of the polar fronts, and also between 100°E and 160°E off the Antarctic coast (Figure 10). The effect of improved

mid-Pliocene palaeogeography seems to be most dominant when comparing SATs obtained from simulations Eoi405_ORB and PlioM1 (Figure 3a). On the contrary for SSTs, the combined effect of all the improved boundary conditions produces a warmer North Atlantic (compare Figs. 8a and b). This implies that the effect of geography on Arctic temperatures is different for atmosphere and ocean realms.

### 3.3 Sea ice

Generally, annual mean sea ice is strongly reduced for both PlioMIP1 and PlioMIP2 mid-Pliocene simulations if compared to the respective PI simulation. The general pattern of mid-Pliocene sea ice is rather similar, in that sea ice cover retreats towards the North Pole (see Figure 11).

For seasonal analysis of sea ice, the definition of seasons is different from the conventional December to February (boreal winter) and June to August (boreal summer) time period. In this study, with regard to sea ice, winter is rather defined as

February to April (FMA), and summer rather as the months from August to October (ASO). According to Howell et al. (2016), these are the three-month-periods in which more than half of the PlioMIP1 ensemble simulations show the highest and lowest mean sea ice extent, respectively. There are slight differences in prescribed orbital parameters, both for PI simulations and the respective mid-Pliocene simulations considered in this study that employ an orbital forcing similar to the Pre-Industrial's. We find that differences in the orbital forcing have no effect on the large scale seasonal pattern of Pre-Industrial Arctic sea ice,

as our results show for respective simulations similar spatial extent for both summer and winter (compare Figure 11a and c, 11b and d). With the mPWP boundary conditions prescribed for PlioMIP1 and PlioMIP2, COSMOS simulates a considerably smaller sea ice extent for the mid-Pliocene with respect to Pre-Industrial simulations (compare Figure 11e, f, g, h to Figure 11a, b, c, d). The most obvious loss of sea ice in the mPWP is present around the Hudson Bay, which is represented as land for mid-Pliocene simulations. In contrast, in our Pre-Industrial simulations, the Hudson Bay is totally ice-free during summer, but

sea ice compactness at this location reaches a maximum during winter. The Canadian Arctic Archipelago is totally free of sea ice for mid-Pliocene simulations with PRISM4 geography. In addition to these trivial changes in sea ice, little or no sea ice is present around the Bering Strait during mid-Pliocene boreal summer.

The combined effect of mid-Pliocene geography, orbit and $CO_2$ (derived from simulations PlioM1 and Eoi400) shows that the mid-Pliocene Arctic ocean is ice free during summer. Sea ice compactness drops below 15% in our PlioM1 simulation, while

for simulation Eoi400 we find that summer sea ice is quasi absent (compare Figure 11f and 11h). If we allow as an additional degree of freedom variation of the orbital configuration within the limits of plausibility during the mPWP, and choose the K1 configuration in simulation Eoi400_K1 rather than the KM5c configuration in PlioM1, Eoi400, and derived simulations, we find a larger sea ice extent and compactness during boreal summer as a result of the K1 orbital configuration (Figure 12). Enhanced warming in the Northern Hemisphere polar regions during summer is simulated for MIS KM5c (Figure 10).





Hence, colder conditions with increased summer sea ice prevail for K1 (see Figure 12b and 12d. Simulation Eoi400 shows considerable amount of boreal winter sea ice around the pole and a gradual reduction towards the continents. This is also the case for simulation Eoi405, despite the 5 ppmv difference between both simulations (compare Figure 11g and 12a). Simulation Eoi400_ORB shows a similar spatial sea ice pattern as simulation Eoi400 (compare Figures 12e, f to 11g, h), which implies

that small changes in orbital forcing between both COSMOS simulations have little or no effect on sea ice. Generally, all simulations with PRISM4 geography produce sea ice with larger spatial extent and compactness during summer months than it is the case for their PRISM3D counterpart PlioM1, irrespective of the specified concentration of atmospheric $CO_2$ (compare Figure 12b,d,f,h with 11f). Therefore, the change in $CO_2$ between PlioMIP1 and PlioMIP2 does not affect large-scale patterns of summer sea ice in the Arctic.

**4   Discussion**

When comparing large scale patterns of mPWP climate simulated by us with COSMOS in the framework of PlioMIP1 and PlioMIP2, we unsurprisingly find that also in our contribution to PlioMIP2 the mPWP offers a glimpse into a climate state that is overall warmer than the conditions humankind is currently experiencing. Noteworthy is that relative warmth in the mPWP is possible with a prescribed $CO_2$ forcing that is actually below the current volume mixing ratio in the atmosphere – 400 ppmv

in PlioMIP2's mPWP, about 407 ppmv for 2018 (Friedlingstein et al., 2019, and one reference therein). Yet, we also infer that updates of the model setup from PlioMIP1 to PlioMIP2 lead to both a global and a regional modulation of the overall warmth simulated for the mPWP with respect to the Pre-Industrial.

One of the main objectives of the PlioMIP is to determine the dominant components of mid-Pliocene warming, derived from the imposed boundary conditions (Haywood et al., 2016). For the COSMOS simulation in response to PlioMIP1 boundary

conditions (simulation PlioM1), the most pronounced warming is evident over areas where changes in albedo and orography have been implemented (Stepanek and Lohmann, 2012). This is also the case for PlioMIP2 simulations with the COSMOS, above all for the PlioMIP2 core simulation Eoi400. However, the increased number of simulations with dedicated sensitivity studies in PlioMIP2, in comparision to the approach in PlioMIP1, that considered only one simulation based on the best knowledge on mPWP boundary conditions (Haywood et al., 2010, 2011), allows a proper inference of the main drivers of

mid-Pliocene warmth. While the study by Stepanek et al. (2020) aims at unravelling the various contributions of PRISM4 boundary conditions to the mPWP climate anomaly as simulated with COSMOS, with this study we employ further sensitivity experiments that go beyond the PlioMIP2 modelling protocol in order to determine the relative contribution of the improved boundary conditions from PlioMIP1 to PlioMIP2. Our aim is to bridge the gap between COSMOS contributions to PlioMIP1 and PlioMIP2, that are based on model setups that differ beyond the discrepancies between boundary conditions outlined in

protocols for PlioMIP1 (Haywood et al., 2010, 2011) and PlioMIP2 (Haywood et al., 2016). Our extended modelling approach allows inference into the contributions of different components to the results achieved with COSMOS in PlioMIP2. Our respective inferences are outlined below.



Generally, we find that the effects of changes in boundary and initial conditions are pronounced in the higher latitudes, while SAT and SST are largely unchanged in the lower latitudes. Hence, for the COSMOS the impact of updates of the modelling methodology from PlioMIP1 to PlioMIP2 (encompassing implementation of PRISM4 boundary conditions with slightly reduced $CO_2$, increased detail of mPWP land sea mask and gateway configuration, resolution of climate-vegetation feedbacks

via the use of the model's dynamic vegetation module) modifies the polar amplification in the simulated mPWP climate. While regionally the PlioMIP2 core simulation Eoi400 is warmer than the respective climate state of PlioMIP1 (simulation PlioM1), in particular where the ice sheet reconstruction is updated in PRISM4, the Arctic is generally cooler in our contribution to PlioMIP2. Lower temperature anomaly in the Arctic imprints on sea ice conditions in high latitudes of the Northern Hemisphere and the global average temperature simulated in PlioMIP2, where a small reduction of the modelled temperature

anomaly is evident from PlioMIP1 to PlioMIP2. Generally, we find that in relation to updated gateways there is a pronounced regional increase of SST in the North Atlantic Ocean, while reduced carbon dioxide leads, as expected, to an overall cooling of the mPWP climate in COSMOS simulations of PlioMIP2. Yet, a comparison of SAT simulated in the framework of a sensitivity study, where the PlioMIP2 core simulation Eoi400 is repeated with the higher $CO_2$ forcing employed in PlioMIP1 simulation PlioM1, reveals that the impact of radiative forcing by increased $CO_2$ is hemispherically dependent. In the PlioMIP2 model

setup with increased $CO_2$ (as in PlioMIP1), the North-Atlantic and large parts of Mediterranean and Arctic are actually cooler than in the standard PlioMIP2 model setup with 400 ppmv $CO_2$. In contrast, for Southern Hemisphere, North Atlantic equatorward of 30°N, and most of the North Pacific, a model setup with the higher PlioMIP1 $CO_2$ forcing would indeed lead to a warmer mPWP state than what is simulated in our PlioMIP2 core simulation Eoi400. Our $CO_2$ sensitivity study within the framework of PlioMIP1 and PlioMIP2 shows that a difference of 5 ppmv between both phases of PlioMIP causes apprecia-

ble changes in SAT over land and oceans. The increase in $CO_2$ produces warmer oceans in the PRISM4-based model setup, except for parts of the North Atlantic and the Arctic as outlined above. As in the PRISM4 COSMOS model setup the effect of increased $CO_2$ is opposite to the pronounced mPWP warming simulated in the North Atlantic, that is found in PlioMIP2 simulation Eoi400 in comparison to PlioMIP1 simulation PlioM1, we conclude that the warmer North Atlantic simulated by us in PlioMIP2 is not influenced so much by changes in greenhouse forcing as it is by the collective contribution of all boundary

conditions. As a matter of fact, the effect that relatively small changes in $CO_2$ have in the PlioMIP2 mPWP setup of COSMOS on the temperature of the North Atlantic Ocean leads us to the inference that the gateway effect on North Atlantic Ocean warmth is modulated in our model by greenhouse forcing, and that our PlioMIP2 model setup does not provide a final statement with regard to the strength of the gateway effect that may have impacted on the mPWP temperature signals interpreted from the geologic recorder. Other model configurations with altered greenhouse gas forcing may well produce a larger tem-

perature anomaly than the COSMOS PlioMIP2 core simulation Eoi400. If we extend our focus beyond model performance within PlioMIP2, then we find that a PlioMIP2 COSMOS simulation with the higher volume mixing ratio of 405 ppmv of $CO_2$ is actually in better agreement with the PlioMIP1 simulation PlioM1 in parts of the North Atlantic Ocean (where that simulation suggests colder SST than the PlioMIP2 core simulation Eoi400). Consequently, our results show that a slightly higher $CO_2$ forcing provides larger disagreement between modelled and reconstructed SST, as both model setups of PlioMIP1

and PlioMIP2 are, in comparison with temperature data derived from proxy records, too cold.





Changes in modern-day orbital forcing between COSMOS PlioMIP1 and PlioMIP2 model setups and the corresponding mid-Pliocene simulations overall promote polar amplification in the former, with the exception of mid- to high-latitude continental regions and parts of the North Atlantic Ocean. In particular, parts of the central Arctic are slightly cooler with the updated orbital forcing in PlioMIP2, while parts of the North Atlantic Ocean are warmer. Beyond the impact of details of the modern-like

Pre-Industrial orbital forcing in our PlioMIP model setups, orbital forcing can play a major role in achieving warmer or colder simulated mid-Pliocene conditions. It can lead to significant annual and seasonal variations (Prescott et al., 2014). Therefore, prescribing modern day orbital parameters for a mid-Pliocene simulation, and comparing the results with proxy data averaged over multiple time-slices within the mid-Pliocene, could increase the cases of data and model discords (Dowsett et al., 2013). Sensitivity tests of PlioMIP1 ensemble models, reported by Salzmann et al. (2013), identified insufficient temporal constraints

hampering the accurate configuration of model boundary conditions as an important factor impacting on data-model discrepancies. Prior to the start of PlioMIP2, a more defined orbital time slice has been suggested to allow a robust evaluation of present climate models to predict warm climates (Prescott et al., 2014). Thus, within the framework of PlioMIP2, a more defined orbital time slice (MIS KM5c) has been specified (Haywood et al., 2016). Even though it reflects a clear signature of MIS KM5c when compared with K1 in the North Atlantic, reconstructions which are available for comparison still reflect signals of multiple

time slices within the mid-Pliocene. This could lead to increased cases of model-data mismatch, as our results demonstrate pronounced annual and seasonal variation in SATs and SSTs between two different orbital time-slices (K1 and KM5c) of the mid-Pliocene, although with a more pronounced signature of the KM5c climate state.

We also find that mid-Pliocene orography and ocean gateways contribute more relative to the location where they are applied. Due to reduction in ice sheet extent, orography over Greenland has been lower by more than 1500 m during the mid-Pliocene

relative to present day (Yan et al., 2013). The COSMOS mid-Pliocene simulation with PlioMIP1 boundary conditions shows that warming related to elevation reduction implied by the reconstructed lower than present Greenland Ice Sheet contributes to about 80% of the total warming at this location when compared with the Pre-Industrial (Stepanek and Lohmann, 2012). Assuming a lapse rate of 6.5 K per elevation change of 1000 m, the PlioMIP2 mid-Pliocene core simulation shows that orography contributes 75% to the mid-Pliocene warming over Greenland, with respect to the prescribed Pre-Industrial simulation.

The PlioMIP1 model ensemble showed a diverse picture of mid-Pliocene AMOC with respect to Pre-Industrial. Only a slight increase of meridional ocean heat transport was found by a small number of models, and where this occured it created only a small increase in meridional ocean heat transport that would contribute to warming in the North Atlantic, while a consistent increase of AMOC and the related meridional ocean heat transport lacks in the PlioMIP1 model ensemble (Zhang et al., 2013). Even in the COSMOS model, where both AMOC and meridional heat transport are slightly enhanced in PlioMIP1, the sim-

ulated warming in the North Atlantic Ocean does not reproduce the relative warmth indicated by the available mid-Pliocene SST reconstruction by Dowsett et al. (2013). In an attempt to simulate enhanced mid-Pliocene AMOC, studies suggested that closure of the Bering Strait would lead to enhanced AMOC (Hu et al., 2015). Furthermore, it has been stated that the modification of certain sea floor features could increase the strength of AMOC (Brierley and Fedorov, 2016; Robinson et al., 2011). With respect to the climate of the mPWP, Otto-Bliesner et al. (2016) have explored the climatic effect of adjustments of Bering

Strait, Canadian Arctic Archipelago, and Hudson Bay, towards their respective states during the Pliocene, on simulated SST.





With changes in prescribed geography for PlioMIP2 mPWP simulations, that incorporate the assumed modifications of ocean gateways in the mPWP with respect to modern, we find that COSMOS indeed simulates an enhanced AMOC for PlioMIP2 with respect to PlioMIP1. Enhanced AMOC relates in COSMOS also to a warmer North Atlantic surface ocean in the PlioMIP2 simulation Eoi400. Consequently, the PlioMIP2 mPWP simulation with adjusted (i.e. closed) Bering Strait reduces the model-

data mismatch found by Dowsett et al. (2013) that is particularly pronounced in the North Atlantic Ocean.

Yet, we acknowledge that PRISM4-conform ocean gateways cannot fully remove the observed model data mismatch in COSMOS. Furthermore, we find that in the region of 30°N and 60°N in the North Atlantic Ocean, where PlioMIP2 simulation Eoi400 is particularly warmer than PlioMIP1 simulation PlioM1, and the model-data mismatch is therefore reduced, SST is also suszeptible to relatively small variations in the forcing of $CO_2$ and details of the Pre-Industrial orbital configuration. It

is therefore particularly challenging to unravel the various contributions to SST increase and to quantify the extent to which gateway configuration is the cause for increased North Atlantic Ocean temperatures. In particular with regard to variations of the Pre-Industrial orbital configuration, we note that the extremely small amplitude of the variation of the orbital forcing suggests that at least a part of the SST effect that we derive from the simulations is caused by internal variability in the climate system rather than an imprint of the modified model setup. Although large parts of the anomaly are found to be statistically

significant, we cannot exclude the possibility that a part of the simulated SST pattern is not a stable climate feature, but rather an overprint of slow modes of climate variability, whose periodicity is beyond the analysis period of 100 model years.

From our SAT analyses we find that PRISM4 palaeogeography seems to be the dominant factor driving the mid-Pliocene warmth. This is shown by comparing the mid-Pliocene simulation with PRISM4 palaeogeography and all other boundary

conditions as in PlioMIP1 (our simulation Eoi405_ORB) with the PlioMIP1 coupled ocean-atmosphere experiment – with the result that the same SAT patterns emerge as generated by the total effect of switching from PlioMIP1 to PlioMIP2 setup (derived from the comparision PlioM1 vs Eoi400). Similar analysis for SST shows that PRISM4 palaeogeography is not the only factor producing a warmer North Atlantic, but the combined effect of PlioMIP2 boundary conditions. Donnadieu et al. (2006) demonstrate that land and ocean climates are affected differently by changes in palaeogeography. Within the framework

of PlioMIP2, PRISM4 palaeogeography prevents fresher sea water from the Pacific from reaching the North Atlantic due to the closure of the Bering Strait, having significant influence on SSTs in this region.

If we turn to the effect of alternative mPWP forcing on climate simulated in the COSMOS, then seasonal analysis of our simulations shows that sea ice and SST are not only sensitive to prescribed $CO_2$. Orbital forcing and the state of imposed paleogeography can also influence simulated sea ice extent and compactness. Reduced spatial extent and thickness is evident

in particular during boreal summer for simulations with KM5c orbit with respect to simulations that are forced by K1 orbit. Simulations with orbital configurations of K1 show colder high latitudes in summer and warmer mid-to-high latitudes, and thus impact on simulated sea ice in the Arctic. This is attributed to high eccentricity estimated for the K1, leading to a more elliptical orbit around the sun which in turn affects the distribution of insolation for different seasons (Fischer and Jungclaus, 2011). While COSMOS PlioMIP1 simulation PlioM1 shows close to ice free conditions in the Arctic during boreal summer,

PlioMIP2 core simulation Eoi400 is characterized by – relatively – increased summer sea ice conditions in the Arctic Ocean



– albeit sea ice cover in this simulation being still low in comparison to, for example, the Pre-Industrial control state PI_2. Our exercise in considering alternative orbital configurations, that are equally plausible for the mPWP as is the reference orbit KM5c, reveils that we can create with the COSMOS a third realisation of mPWP Arctic summer sea ice that differs from both the PlioMIP1 and PlioMIP2 COSMOS mPWP sea ice states. Albeit overall the K1 orbit provides larger total incoming solar

shortwave radiation than the Earth receives during modern times (Prescott et al., 2014), under its influence Arctic summer sea ice is enhanced in comparison to the PlioMIP2 core simulation. This effect is even stronger with respect to the mPWP state simulated with COSMOS in PlioMIP1. These findings highlight the degree of variability that is present in the simulation of Arctic Ocean sea ice conditions across the mPWP – even if only one climate model is considered, that is exposed only to those modifications of boundary conditions that are considered to be within the range of possibility during the mPWP. Consequently,

variability of Arctic sea ice in our study reminds us of the temporal variability that sea ice may have been subject to across the Pliocene epoch. Variability of Arctic sea ice, more precisely its temporal evolution, has been linked to the development of the Greenland Ice Sheet (Clotten et al., 2019). Therefore, we highlight the scientific merit that may arise from studying both inter-model- and time-variability of Arctic sea ice across the mPWP. This may be done, for example, in the framework of upcoming iterations of PlioMIP, and may lead us towards a better understanding of the climate transition from Cenozoic "greenhouse" to

Pleistocene "icehouse". We highlight that in particular the study of inter-model variability of sea ice evolution across multiple orbital configurations during the mPWP, that is beyond the framework of our publication, should be pursued in future work. While simulations of MIS KM5c employ the orbital configuration of present day, which has been demonstrated to simulate ice free summer for the mid-Pliocene, there is an appreciable degree of inter-model spread that is at a similar amplitude as uncertainty in proxy-based reconstructions of mPWP Arctic sea ice extent (Howell et al., 2016). Variability in simulated sea

ice extent and thickness within the PlioMIP1 ensemble is reported to be the effect of differences in sea ice components of each model (Howell et al., 2016). Consequently, the various climate models employed in PlioMIP, that feature very different types of sea ice components, may tell very different stories about the evolution of Arctic sea ice during the Pliocene. Yet, our results highlight once more that even within one climate model, the COSMOS in our case, many realisations of Pliocene Arctic sea ice are possible. Their presence or absence in the climate simulation may be triggered by relatively small (in the context of

the geologic history of the recent Cenozoic) modifications of the model setup, that encompass in our study the switch from the PRISM3D to the PRISM4 paleogeographic reconstruction, small modifications in orbital forcing and greenhouse gases, as well as the phase space of equally realistic orbital configurations during the mPWP. Any of these changes opens up another window of study of Pliocene Arctic sea ice variability and of its impact on Cenozoic climate evolution.

We conclude our discussion by noting that results of this study suggest that there is an appreciable amplitude of internal

variability inherent in mPWP climate simulations produced with the COSMOS – potentially this is also the case for other models, which could be tested in the future. One of our findings is that relatively small variations of the prescribed climate forcings, that are represented in our study by variations of greenhouse gas concentrations in the order of a few ppmv of $CO_2$ and very minor modifications of an otherwise Pre-Industrial orbital configuration, can lead to small, but non-negligible, mPWP climate anomalies. While we did not explicitly study the reasons behind this finding, the small amplitude of the overall forcing

anomaly suggests that the linked variation of the mPWP climate anomaly, computed over a time period of 100 model years, is





likely at least partly a result of modulation of internal variablity in the model that is triggered by alteration of the model forcing. While PlioMIP2 already made a major leap forward by increasing the analysis time period, from 30 years in PlioMIP1 to 100 years in PlioMIP2, this finding highlights that in future intercomparisons it may be worthwhile to further increase the analysis time period towards multicentennial time scales. Our suggestion is motivated by the observation that beyond the common short

term climate variability, that occurs for example in the form of the El Nino Southern Oscillation (Christensen et al., 2013, see their Figure 14.13) and North Atlantic Oscillation (Hurrell, 1995, see their Figure 1A), there are certain modes of internal variability that act at ocean basin scale with much slower periodicity. If the PlioMIP analysis period were at multicentennial time scales, such slow modes of internal variability would be more likely suppressed by averaging over multiple realizations of climate patterns that are related to opposite phases of these modes, hence leading to a clearer emergence of a mean (that

means, predominant across time periods of centuries) mPWP climate state.

## 5   Summary and conclusions

In this study, we compared the results of mid-Pliocene simulations within the frameworks of PlioMIP1 and PlioMIP2, that have been carried out with the COSMOS at the Alfred Wegener Institute in Germany. We have tested to which degree changes between the COSMOS model setups of PlioMIP1 and PlioMIP2 are able to create different realizations of simulated mPWP

climate, and how different these alternative mPWP climate states are from PlioMIP2 core simulation Eoi400 and PlioMIP1 simulation PlioM1, both produced with the same climate model as employed in this study. Overall, we find that the global patterns of mPWP climate as expressed in SST and SAT are similar in PlioMIP2 and PlioMIP1. Yet, in PlioMIP2 we simulate certain differences that impact in particular sea ice conditions in the Arctic and the amplitude of mPWP temerpature anomaly in the North Atlantic Ocean. Beyond small changes of the Pre-Industrial orbit reference configuration in COSMOS and the

modification of $CO_2$, that arises from the transition of PRISM3D to PRISM4 boundary conditions, we carried out sensitivity experiments that address the climatic impact of plausible orbital configurations across the mPWP (MIS KM5c and K1) to assess variability across different discrete time-slices within the mid-Pliocene warm period, using boundary conditions prescribed for PlioMIP2.

From our results, we conclude that palaeogeography is the dominant component of the imposed mid-Pliocene boundary con-

ditions, influencing the climate across the warm period. Its effect on atmosphere and ocean differs as seen in the different pattern of warming and cooling achieved for SAT and SST. We also conclude that the response of atmospheric and oceanic variables differ distinctly when exposed to the same geography. The difference in $CO_2$ between PlioMIP1 and PlioMIP2 simulations does not change the general impression of large scale mPWP climate patterns, but produces warmer oceans especially in high latitudes of the Northern Hemisphere. A similar effect we find if we produce simulations with slight alterations of the

Pre-Industrial surrogate of KM5c orbital forcing. The impact of a small amplitude of change in orbital forcing leads us to the suggestion that the simulated PlioMIP climate state may be subject to appreciable internal variability that – while not changing the overall impression on the characteristics of simulated mPWP climate – provides regional patterns of SST and SAT that are worthwhile to be studied in future phases of PlioMIP based on a prolonged multicentennial analysis period.



Furthermore, the newly prescribed PlioMIP2 boundary conditions play a significant role in achieving warmer North Atlantic SSTs, that were modelled too cold during PlioMIP1, if compared to reconstructions. On the other hand, the PlioMIP2 mPWP reference state simulated with COSMOS (simulation Eoi400) is globally slightly cooler, in agreement with reduced concentrations of $CO_2$, and also expresses itself via reduced warmth and resultantly increased summer sea ice conditions in the Arctic

Ocean. When considering various modifications of mPWP boundary conditions in our simulations, we find that there is the potential for pronounced variability of Arctic summer sea ice conditions during the mPWP. Enhanced AMOC in PlioMIP2 is achieved through the combined effect of PlioMIP2 boundary conditions, contributing to increased meridional transport of warm water from low to mid latitudes in the North Atlantic. We note that details of the model forcing, in particular the exact level of carbon dioxide and the realization of the KM5c orbit surrogate, have an impact on the amplitude of North Atlantic

Ocean warming in comparison to results derived with COSMOS in PlioMIP1. With respect to Pre-Industrial conditions, extended equatorial warm pool, shown in COSMOS simulation PlioM1 for PlioMIP1, is also present in our PlioMIP2 simulation Eoi400 with similar spatial pattern and extent. This is in line with our finding that the effect of changes in mid-Pliocene boundary conditions from PlioMIP1 to PlioMIP2 is minor in low latitudes, and more pronounced in high-latitudes. The magnitude of Arctic warming simulated for PlioMIP1 is reduced in our PlioMIP2 simulation, while increased warming over Greenland

and Antarctica is a direct effect of changes in reconstructed orography and albedo from PlioMIP1 to PlioMIP2.

Employing a different plausible mPWP orbital forcing (MIS K1) in simulating the mid-Pliocene, and comparing the results to those derived based on the standard Pre-Industrial orbital configuration that is used as a surrogate for the MIS KM5c, reveals pronounced climate anomalies that may express themselves via both warming and cooling at various spatial scales – the former mostly at high latitudes, and in particular in the Arctic Ocean, the latter widespread in low- to mid latitudes. Comparison

between climate states simulated for both the KM5c and K1 time-slices shows annual and seasonal variations, with RMSD between either one of the modelled North Atlantic SSTs for these time slices on one side, and reconstructions of mPWP SSTs on the other side, showing a better agreement between the geologic record and the simulation for MIS KM5c (Eoi400) in contrast to MIS K1. This confirms a more pronounced signal of KM5c climate state in the reconstruction. Yet, in line with statements by previous authors (Dowsett et al., 2016; Haywood et al., 2016; Prescott et al., 2014; Salzmann et al., 2013),further steps

should be taken to reconstruct mid-Pliocene climate within the KM5c time-slice, as this will reduce signals and interferences that could be caused by averaging over multiple isotope excursion periods, when comparing with modelled data.

*Data availability.* The COSMOS outputs utilized for this study, i.e. selected model output of simulations Eoi405, Eoi400_ORB, simulation Eoi405_ORB, and simulation Eoi400_K1, will be made available via PANGAEA by the time the review phase of the manuscript has been concluded. Model output related to PlioMIP1 (simulations PI_1 and PlioM1) and PlioMIP2 (simulation PI_2, named E280 in the framework of the study by Stepanek et al. (2020), and simulation Eoi400) are used in the framework of PlioMIP1 and PlioMIP2.





*Author contributions.* E. S., C. S. and G. L. designed the study concept and structure of the work, E.S and C.S implemented model runs, E.S analysed the resulting data and wrote the manuscript, incorporating comments from co-authors. Correspondence should be addressed to E.S.





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



**Table 1.** Detailed description of experiments including category of orbital configuration, based on two different Pre-Industrial (PI) and one mPWP-specific setting, as well as specific parameter values and applied geography.

| Category | Exp. ID | Geography | CO$_2$ (ppmv) | CH$_4$ (ppbv) | N$_2$O (ppbv) | Orbital configuration | Eccentricity | Obliquity (°) | Perihelion (°) |
|---|---|---|---|---|---|---|---|---|---|
| **Pre-Industrial** | PI_1 | ETOPO5 | 280 | 760 | 270 | PI 1 | 0.0167240 | 23.446000 | 282.04000 |
| | PI_2 | ETOPO5 | 280 | 808 | 273 | PI 2 | 0.0167643 | 23.459277 | 280.32687 |
| **PlioMIP1** | PlioM1 | PRISM3D | 405 | 760 | 270 | PI 1 | 0.0167240 | 23.446000 | 282.04000 |
| **PlioMIP2** | Eoi400 | PRISM4 | 400 | 808 | 273 | PI 2 | 0.0167643 | 23.459277 | 280.32687 |
| | Eoi405 | PRISM4 | 405 | 808 | 273 | PI 2 | 0.0167643 | 23.459277 | 280.32687 |
| | Eoi400_ORB | PRISM4 | 400 | 808 | 273 | PI 1 | 0.0167240 | 23.446000 | 282.04000 |
| | Eoi405_ORB | PRISM4 | 405 | 808 | 273 | PI 1 | 0.0167240 | 23.446000 | 282.04000 |
| **PlioMIP2_K1** | Eoi400_K1 | PRISM4 | 400 | 808 | 273 | MIS K1 | 0.0536210 | 23.011620 | 223.15315 |



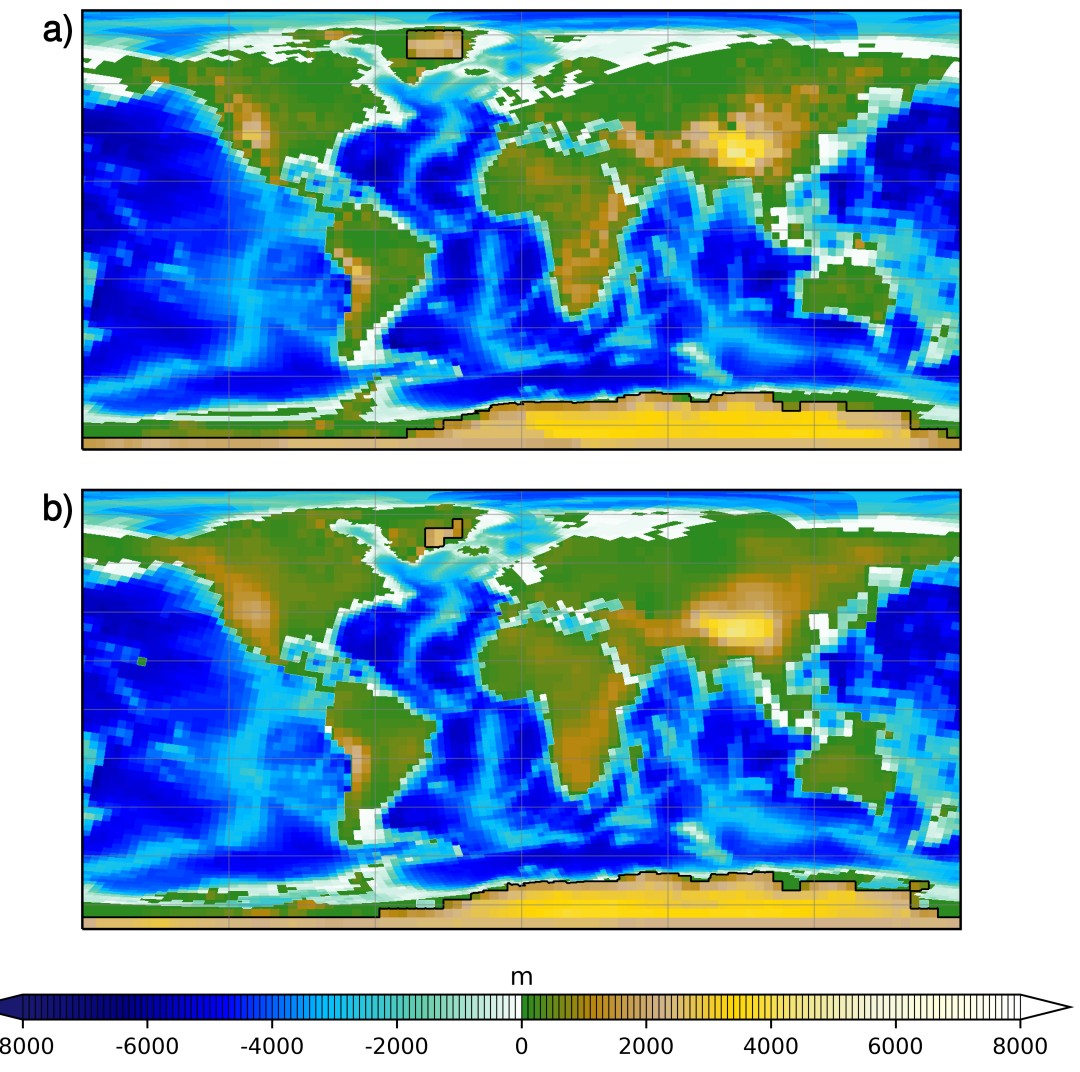

**Figure 1.** (a) PRISM3D land-sea mask implemented in COSMOS simulations for PlioMIP1 based on data provided by Haywood et al. (2010), assuming presence of an open Bering Strait and Canadian Arctic Archipelago. (b) PRISM4 land-sea mask implemented in COSMOS simulations for PlioMIP2 with closed Bering Strait and Canadian Arctic Archipelago as described in the PlioMIP2 protocol (Haywood et al., 2016; Dowsett et al., 2016). Color shading depicts the prescribed land orography and ocean bathymetry (m) for PlioMIP1 and PlioMIP2, respectively. Black isolines depict prescribed mid-Pliocene ice sheets for the respective phases of PlioMIP. Differences over Antarctica are due to the switch from employing a modern land sea mask with minor modifications towards Pliocene conditions in the COSMOS PlioMIP1 simulation, and employing a full Pliocene representation of Antarctic geography in COSMOS for PlioMIP2.







**(a)**

**(b)**

**Figure 2.** Annual mean SAT (K) anomalies between core mid-Pliocene simulations for PlioMIP1 and PlioMIP2, with their respective Pre-Industrial simulations (PI_1 and PI_2). **(a)** Climatological anomaly that has been calculated over 100 years for PlioMIP1; **(b)** as a, but for Eoi400. Stipples (black dots) show regions of statistically insignificant differences.





**Figure 3.** Annual mean SAT (K) anomalies between mid-Pliocene simulations with varying boundary conditions. **(a)** Eoi405_ORB - PlioM1, which shows the contribution of changes in mid-Pliocene palaeogeography between PlioMIP1 and PlioMIP2; **(b)** Eoi400 - PlioM1. Comparison of mid-Pliocene simulations of both PlioMIP phases, incorporating all the changes from PlioMIP1 to PlioMIP2.





**Figure 4.** Annual mean SAT (K) anomalies between mid-Pliocene simulations with varying boundary conditions. **(a)** Eoi405 - Eoi400, showing anomalies due to changes in mid-Pliocene $CO_2$ from 405 to 400 ppmv as utilized for PlioMIP1 and PlioMIP2 respectively, while **(b)** Eoi400_ORB - Eoi400, show variations caused by changes in mid-Pliocene orbit utilized in COSMOS simulations for PlioMIP1 and PlioMIP2.





**Figure 5.** Annual and seasonal mean SAT anomalies, due to change in mid-Pliocene orbital forcing between Marine Isotope Stages K1 and KM5c, calculated from Eoi400_K1 - Eoi400. Shown are **(a)** annual mean SAT (K), **(b)** boreal summer (JJA) and **(c)** boreal winter (DJF).

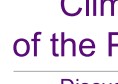
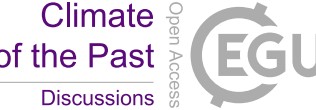

**Figure 6.** Comparison between mid-Pliocene SST (°C) anomalies, calculated from 100 years of model output, with respect to Pre-Industrial control simulations as obtained from COSMOS **(a)** PlioMIP1, and **(b)** PlioMIP2. Black isolines show the expansion of equatorial warm pool in the mid-Pliocene state while, the green isolines depict absolute annual mean sea ice cover in the mid-Pliocene state. The contour interval for the sea ice isolines is 15%.





**Figure 7.** COSMOS simulated mid-Pliocene annual mean AMOC in Sv for **(a)** PlioMIP1 (simulation PlioM1) and **(b)** PlioMIP2 (simulation Eoi400). Overturning rates are time averages that have been calculated from 100-year model outputs. Positive values represent a clockwise circulation.





**Figure 8.** Annual mean SST (°C) anomalies between mid-Pliocene simulations with varying boundary and initial conditions. **(a)** Eoi405_ORB - PlioM1, which shows the contribution of changes in mid-Pliocene palaeogeography between PlioMIP1 and PlioMIP2; **(b)** Eoi400 - PlioM1, comparing mid-Pliocene simulations of both phases of PlioMIP, incorporating all the changes in paleogeography, orbital and greenhouse gas forcing from PlioMIP1 to PlioMIP2.



**Figure 9.** Annual mean SST (°C) anomalies between mid-Pliocene simulations with varying boundary conditions. **(a)** Eoi405 - Eoi400, quantifying anomalies due to changes in mid-Pliocene $CO_2$ from 405 to 400 ppmv, as utilized for PlioMIP1 and PlioMIP2 respectively, while **(b)** Eoi400_ORB - Eoi400, quantifying changes in the (Pre-Industrial, respectively KM5c) orbital configuration as utilized in COSMOS simulations for PlioMIP1 and PlioMIP2.

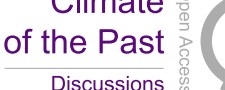

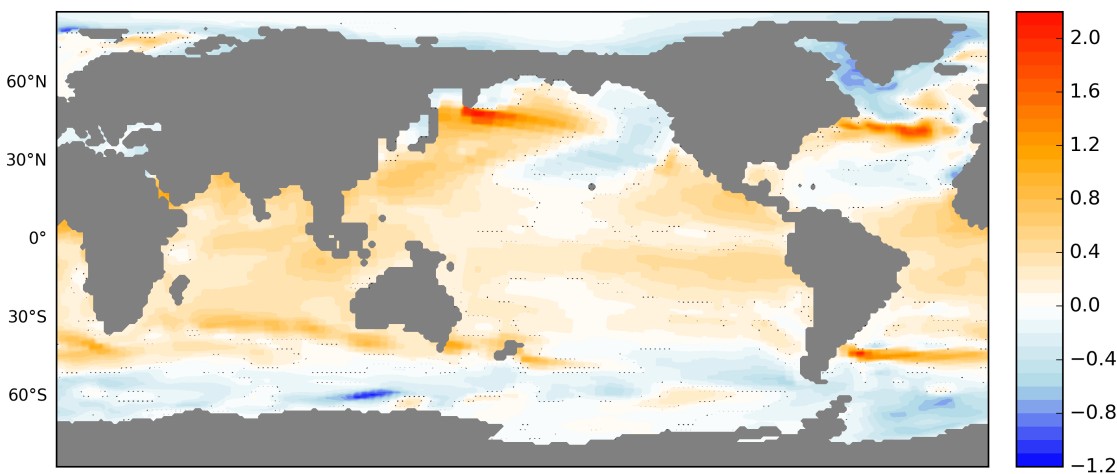

**Figure 10.** Annual mean SST anomalies due to change in mid-Pliocene orbital forcing between Marine Isotope Stages K1 and KM5c, calculated from Eoi400_K1 - Eoi400.

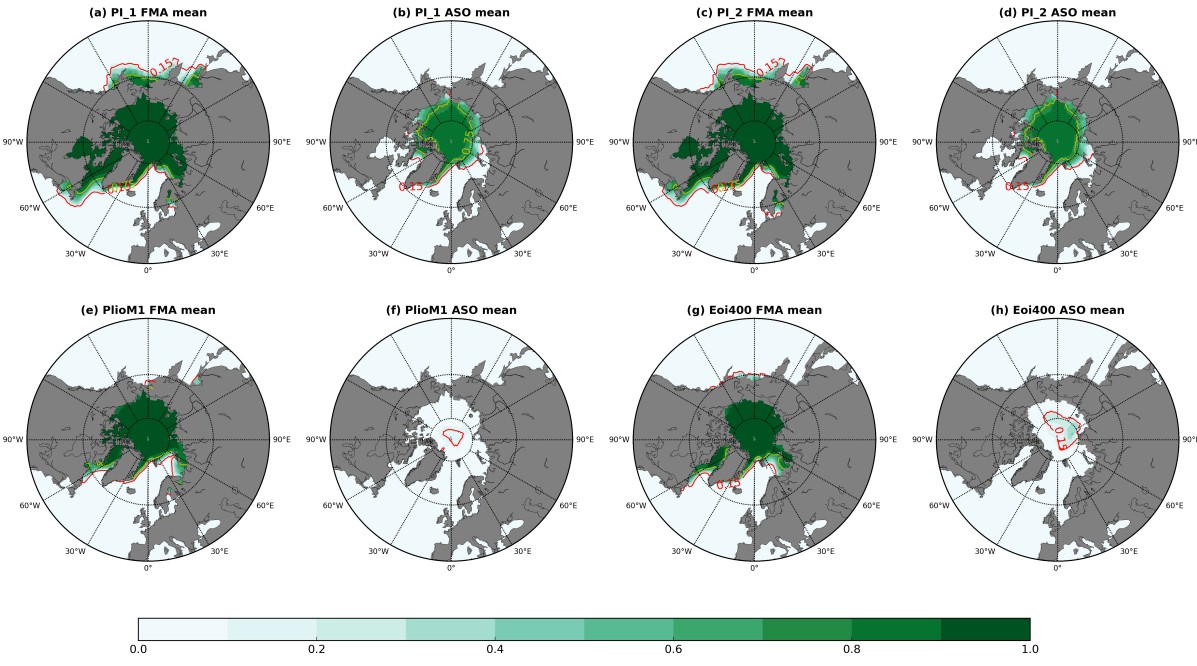

**Figure 11.** Seasonal mean sea ice compactness averaged over 100 model years as simulated by COSMOS for mid-Pliocene and Pre-Industrial runs with response to boundary conditions prescribed for PlioMIP1 and PlioMIP2. (**a**), (**c**), (**e**) and (**g**) show winter (FMA) averages, while (**b**), (**d**), (**f**) and (**h**) show summer (ASO) averages for simulations PI_1, PI_2, PlioM1 and Eoi400, respectively. The red contours indicate the 15% isoline of sea ice cover while yellow contours indicate the 75% isoline of sea ice cover.





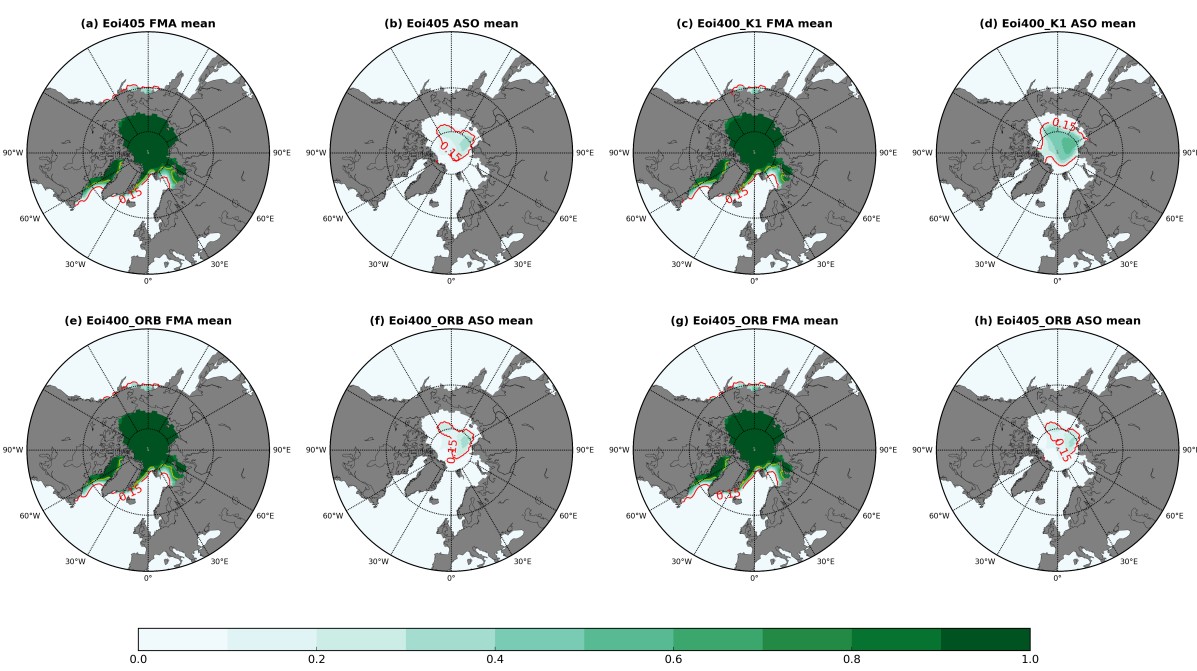

**Figure 12.** Seasonal mean sea ice compactness averaged over 100 model years as simulated by COSMOS for mid-Pliocene simulations with response to changes in PlioMIP's prescribed boundary conditions. **(a)**, **(c)**, **(e)** and **(g)** show winter (FMA) averages, while **(b)**, **(d)**, **(f)** and **(h)** show summer (ASO) averages for simulations Eoi405, Eoi400_K1, Eoi400_ORB and Eoi405_GHG_ORG, respectively. The red contours indicate the 15% isoline of sea ice cover while yellow contours indicate the 75% isoline of sea ice cover.