# Peer review of "Sensitivity of mid-Pliocene climate to changes in orbital forcing, and PlioMIP's boundary conditions"

_Climate of the Past, 2020_

## Referee Comment (RC1) · Anonymous Referee #1 · 3 Mar 2020

General comment : This manuscript presents the results of simulations using the COS-MOS model and different boundary conditions in the frame of PlioMIP. PlioMIP1 and PlioMIP2 simulations are compared, minor differences in the boundary conditions are also investigated (small changes in orbital configuration for the preindustrial, 5 ppm difference between the CO2 prescription from PlioMIP1 to PlioMIP2), and finally a simulation using the orbital configuration of MIS K1 is carried out. The goal is to understand 1/how some minor changes in the boundary conditions, both for the PI and the Pliocene simulations, and 2/ a change of orbit, can impact the results of PlioMIP2.

The scientific question is understandable and the manuscript is relatively easy to follow. The figures are clear. The manuscript is generally well written. Overall it is an interesting contribution to the Special Issue of PlioMIP2 as the scientific question is relevant for this topic. I recommend that it is published after some modifications have been made, notably on one major comment.

I have one major comment on your sensitivity tests to changing CO2 from 405 to 400 ppm, and the large difference in the North Atlantic SST due to this change : as you state in the discussion, this is probably not a 'real' signal. It is, as you state, either due to longwave oceanic variability, or to the fact that one of your simulations is not in equilibrium regarding NADW formation. Could you please check your NADW formation or mixed layer depth for the two simulations concerned (Eoi400 and Eoi405), across the whole integration period ? How long was your integration time for both these simulations ?

In any case, because you state yourself that this cold SST anomaly is certainly an artefact, you have to get rid of this artefact otherwise you cannot discuss the results of this sensitivity test. You can try to overcome this artefact by integrating each simulation on 200 years, or by continuing your simulations if they were in fact not in equilibrium regarding NADW.

To make the manuscript easier to follow, in my opinion, the authors should refer more frequently to the name of each simulation they're describing rather than describing which simulation they are talking about (i.e. writing 'PlioM1' rather than 'the PlioMIP1 simulation' That would make the manuscript more concise and easier to follow (at least for me). Also, in the Discussion section, please refer to the figures.

I also think that for the sake of clarity and answering more properly to the scientific question raised, the Results section could have been organized by forcing, rather than by climatic variable (i.e impact of changing CO2, impact of MIS K1 orbit, etc. rather than 'SST', 'SAT' etc.).

Curiously you do not show a single precipitation map. Did you look at them and see

that only very minor changes appeared ? Please explain the reasons for this choice, as precipitation is an important component of climate, especially at low latitudes.

Specific comments

Abstract

Do not detail the minor changes in boundary conditions here. (page 1, line 5 to 10)

It looks like the abstract was written before the paper was really finished. Some of the conclusions of the abstract are in contradiction with the conclusions of the paper, for example, page 2 :

"The difference in prescribed CO2 accounts for 1.1K of warming in the Arctic, leading to an ice-free summer in the PlioMIP1 simulation and a quasi ice-free summer in PlioMIP2 " → where do you get that information ? from figure 4a and figure 9a there are only «0.5°C changes in the Arctic. The big signal is in the North Atlantic, but is probably not robust. Second, you conclude in your conclusions that CO2 change is likely not the cause of the changes between PlioMIP2 and PlioMIP1, the factor of change being mostly paleogeography changes.

So please rewrite the abstract carefully.

Page 2, line 25 Consistency : use mid-Pliocene not Mid-Pliocene and Plio-Pleistocene instead of Pleistocene-Pliocene

Page 3, line 18 "all model data. . .". This sentence is confusing to me, and could lead a non-specialist reader to think that you feed the model with the same data that you compare it to. Please be more specific like "Boundary conditions of the Pliocene simulations are directly based on PRISM"

Page 5 line 10 "we create model setups where the prescribed PlioMIP2 model setup. . ." → we carry out an additional simulation using PlioMIP2 boundary conditions except for orbital configuration which is representative of MIS K1.

Experimental design

Please provide integration length for all simulations.

Results

As I said in the General comments I think this Section would be clearer if it was organized in terms of forcing rather than in terms of variable. Also, please provide temperatures in degrees Celsius rather than in K, because few people speak in K and your figures are in degrees C. Please refer to the figures whenever necessary, it's not always the case in particular in the SAT section.

Page 1, lines 10 to 20 : comparing the SST dataset of Dowsett et al., 2013 to PlioMIP2 results is irrelevant. The Dowsett 2013 dataset includes data spread over a large amount of time, and peak-averaging. You have to compare to the new dataset by Foley and Dowsett 2020. Also, you here speak of RMSD between this dataset and several simulations but you should provide a table for the reader to refer to.

Discussion

Please also refer to the figures whenever necessary in this section.

Page 13, line 1 "effects of changes in boundary and initial conditions". I did not see that you had changed initial conditions, and if you have an effect from a change in initial conditions that means your simulation has not reached equilibrium, doesn't it ?

Please revise the discussion regarding the effect of 5 ppm $CO_2$ change on the SST after you have found a way to remove the artefact of the cold signal in the North Atlantic, either by averaging on longer time period, or by running the model to full equilibrium regarding the NADW formation.

Page 16 discussion on sea-ice : "may tell very different stories about the evolution of sea-ice". Certainly, different models lead to different sea-ice simulations. However, what I conclude from your results is, a small change in forcing leads to small changes

in sea-ice, but the big story in the same in all you PlioMIP simulations. With COSMOS, in Pliocene conditions, you have strongly reduced sea ice with almost sea-ice free summers in PlioM1, Eoi400, Eoi405, Eoi400_ORB and Eoi405_ORB, and a remarquably similar winter sea-ice extent for all these simulations and Eoi400_K1. Slightly more Arctic sea ice in summer with Eoi400_K1. To me, all these simulations, except maybe Eoi400_K1 which has slightly more ice, tell the same story of sea-ice. But these changes are anyway much smaller than the precision that sea-ice proxies can provide.

By the way what is sea-ice compactness ? Did you mean sea-ice thickness ? I have never seen sea-ice compactness before.

Conclusions Page 17 Please update the conclusions regarding the effect of 5 ppm $CO_2$ change in the North Atlantic, according to the comments above.

I hope that my comments are helpful to the authors. Sincerely.

---

## Referee Comment (RC2) · Anonymous Referee #2 · 1 May 2020

general commentsïijŽ This paper identifies and discusses differences in the modelled climate that occur between the COSMOS simulations contributed to PlioMIP1 and PlioMIP2, and studies the effect of orbits warmer than the configuration applied for PlioMIP2. The sensitivity of mid-Pliocene climate (SST, SAT and the distribution of sea ice) to minor differences in PlioMIP's boundary conditions are investigated. The paper is a significant contribution to the model-model and model-data-comparison within PlioMIP2. The basic conclusion from the paper appears sound and the methods are also generally appropriate and well described. Overall I recommend publication subject to minor revision. 1ïijĽThe title seems to be not so consistence with the introduction. From the title, I think the changes in orbit forcing and PlioMIP's boundary conditions are

all the main topic, even the orbit forcing (basically orbit forcing is one of the PlioMIP's boundary conditions) is more important. However, from the introduction, only the last 5 lines (line 27-31 in page 3) you mention the orbit forcing. And you only did one orbit forcing experiment, that gives me a feeling whether it's necessary to put the orbit forcing in this paper. Maybe only analyzing the impact of the changes in PlioMIP's boundary conditions to mid-Pliocene climate could be better. 2ïijĽAdditionally, I think precipitation and temperature are two basic climate variable. Adding some analyse of the precipitation could be better.

Specific commentsïijŽ 1)Page 2, line 25: delete "and the Pleistocene-Pliocene". You only study the mid-Pliocene period, never investigate anything during the Pleistocene-Pliocene period.   2)Page 5, line 6-8:  the sentence "that is useful. . . . . .that is based. . . . . .", there are two "that is". 3)Page 5, line 19-21: I think there are three major components, not four. 4)Consistency: the authors use "PI" instead of "pre-Industrial", but not all. I think except the first one, the others all use PI could be better. In figure 2, 3, 4, 5 and the main body, the temperature's unit is K; but in figure 6, 8, it is degree C. Maybe all use degree C could be better. 5)Some paragraphs are really too long (e.g. the first paragraph in SAT, the third paragraph in discussion), it's really hard for readers to follow (at least for me). I suggest the authors to divide those long paragraphs into two or three shorter paragraphs. 6)Page 17, line 27-29, the sentence "The difference in CO2 between PlioMIP1 and PlioMIP2 simulations does not change the general impression of large scale mPWP climate patterns, but produces warmer oceans especially in high latitudes of the Northern Hemisphere" is not so clear. I don't know which one produces the warmer oceans, the PlioMIP1 or PlioMIP2? 7)What's the "sea ice compactness" mean? In the results, the authors use "sea ice extent and compactness", but in the discussion, "sea ice extent and compactness" and "sea ice extent and thickness" are all used. Does that mean "compactness" equal to "thickness"? 8)As a reader, I think the discussion is not so clear and logically organized.

technical correctionsïijŽ Missing space, comma and back bracket: page 9, line 17,

use "Hemisphere, we find" instead of "Hemisphere we find"; page 11, line30, use "Figure 11f, h" instead of "Figure 11f and 11h"; page 12, line 1, use "(see Figure 12b, d)" instead of "(Figure 12b and 12d"; page 12, line 8, use "(compare Figure 12b, d, f, h with 11f)" instead of "(compare Figure 12b,d,f,h with 11f)"; page 18, line 24, use " Salzmann et al., 2013), further" instead of " Salzmann et al., 2013),further". There are a lot these kind of mistakes, please check the whole paper carefully.

Please also note the supplement to this comment:
https://www.clim-past-discuss.net/cp-2020-5/cp-2020-5-RC2-supplement.pdf
* * *

---

## Author Comment (AC1) · 29 May 2020

**Reply to comments by anonymous referee #1**

We would like to thank anonymous referee #1 for his/her time and effort in reviewing our manuscript (cp-2020-05). The comments raised in the review are highly appreciated and have helped us to clarify our statements and to further improve our manuscript. In the following, we respond to comments raised in the review.

**MAJOR COMMENT(S)**

\* I have one major comment on your sensitivity tests to changing $CO_2$ from 405 to 400 ppm, and the large difference in the North Atlantic SST due to this change : as you state in the discussion, this is probably not a 'real' signal. It is, as you state, either due to longwave oceanic variability, or to the fact that one of your simulations is not in equilibrium regarding NADW formation. Could you please check your NADW formation or mixed layer depth for the two simulations concerned (Eoi400 and Eoi405), across the whole integration period ? How long was your integration time for both these simulations?

We thank the reviewer for pointing this out, both simulations were integrated for 1500 years as shown in Figure S1.

[Figure]

Figure S1: Time-series of AMOC index across the whole integration period of 1500 years for simulations Eoi400 (red) and Eoi405 (blue). The AMOC index is defined as the maximum in the stream function below 500 m and polewards from 20°N in the North Atlantic, smoothened using 12-year moving average to reduce inter-annual variability. The green shading shows 100-year period used in calculating SST anomalies shown in the manuscript, while the grey shading shows the 100-year period added to the analysis of both simulations to get rid of the cold pool in the North Atlantic. The results and discussion section will be updated in the revised manuscript.

[Figure]

Figure S2 (Figure 9a in manuscript): Annual mean SST (∘C) anomalies between mid-Pliocene simulations Eoi405 and Eoi400, quantifying anomalies due to changes in mid-Pliocene $CO_2$ from 405 to 400 ppmv, as utilized for PlioMIP1 and PlioMIP2 respectively.

** To make the manuscript easier to follow, in my opinion, the authors should refer more frequently to the name of each simulation they're describing rather than describing which simulation they are talking about (i.e. writing 'PlioM1' rather than 'the PlioMIP1 simulation' That would make the manuscript more concise and easier to follow (at least for me). Also, in the Discussion section, please refer to the figures.

We agree with the reviewer that when referring to simulations in some parts of the manuscript, they are described again. We have fixed this by calling the simulations by their ID as shown in Table 1 of the manuscript.

*** I also think that for the sake of clarity and answering more properly to the scientific question raised, the Results section could have been organized by forcing, rather than by climatic variable (i.e impact of changing CO2, impact of MIS K1 orbit, etc. rather than 'SST', 'SAT' etc.).

We have revised the result section and separated it into different subsections according to the contribution of different forcings.
- Impact of Changing $CO_2$
- Effect of MIS K1 orbit on PlioMIP2 simulations
- Contributions of PlioMIP's Palaeogeography

**** Curiously you do not show a single precipitation map. Did you look at them and see that only very minor changes appeared? Please explain the reasons for this choice, as precipitation is an important component of climate, especially at low latitudes.

The main idea going into this study was to infer the major driver of the mid-Pliocene warmth, hence the sensitivity studies by changing different boundary conditions and analyzing SAT and SST. According to your suggestion, We have added some precipitation analyses showing the difference between PlioMIP1 and PlioMIP2 core simulations in this supplement (Figure S3). Does the editor suggest we add any of this to the revised manuscript?

[Figure]

Figure S3: (a) Annual mean anomalies of precipitation (mm/day) between Eoi400 and PlioM1, while (b) shows zonal averages, where the solid black line shows Eoi400 while the dashed line denotes PlioM1. Furthermore, the red line represents of the anomaly between both simulations.

**SPECIFIC COMMENTS**

**Abstract**
1. Do not detail the minor changes in boundary conditions here. (page 1, line 5 to 10)

We have removed the details of minor changes in boundary conditions from the abstract.

2. It looks like the abstract was written before the paper was really finished. Some of the conclusions of the abstract are in contradiction with the conclusions of the paper, for example, page 2 : "The difference in prescribed $CO_2$ accounts for 1.1K of warming in the Arctic, leading to an ice-free summer in the PlioMIP1 simulation and a quasi ice-free summer in PlioMIP2" → where do you get that information ? from figure 4a and figure 9a there are only «0.5◦ C changes in the Arctic. The big signal is in the North Atlantic, but is probably not robust. Second, you conclude in your conclusions that $CO_2$ change is likely not the cause of the changes between PlioMIP2 and PlioMIP1, the factor of change being mostly paleogeography changes.

We agree with the reviewer, that the initial draft of the manuscript compared the core PlioMIP1 and PlioMIP2 simulations to account for $CO_2$ difference but we later realized that other boundary conditions could also contribute in this respect, hence the implementation of Eoi405. We have made the necessary correction and the abstract has been re-written based on the comments.

3. Page 2, line 25 Consistency : use mid-Pliocene not Mid-Pliocene and Plio-Pleistocene instead of Pleistocene-Pliocene

Mid-Pliocene changed to mid-Pliocene and Pleistocene-Pliocene deleted based on RC2.

**Experimental design**
1. Please provide integration length for all simulations.

We have added the integration length for all simulations to the experimental design section of the manuscript.

**Results**
1. As I said in the General comments I think this Section would be clearer if it was organized in terms of forcing rather than in terms of variable. Also, please provide temperatures in degrees Celsius rather than in K, because few people speak in K and your figures are in degrees C. Please refer to the figures whenever necessary, it's not always the case in particular in the SAT section.

- The results section has been re-organized into different subsections according to the contribution of different forcings.
- Temperature unit K has been changed to degree Celsius to ensure consistency.
- We have referred to figures whenever necessary.

2. Page 1, lines 10 to 20: comparing the SST dataset of Dowsett et al., 2013 to PlioMIP2 results is irrelevant. The Dowsett 2013 dataset includes data spread over a large amount of time, and peak-averaging. You have to compare to the new dataset by Foley and Dowsett 2020. Also, you here

speak of RMSD between this dataset and several simulations but you should provide a table for the reader to refer to.

We thank the reviewer for pointing this out. The SST dataset of Foley and Dowsett (2019) was not available during the preparation of this manuscript. However, we have now compared our results with the reconstructions of Foley and Dowsett (2019) and the result is shown in the table below. Further comparison with the reconstructions of McClymont et al., (2020) will be added in the revised manuscript.

Table S1. Root mean square deviation between Atlantic sea surface temperatures of selected simulations and the alkenone $U^k_{37}$ based reconstructions by Foley and Dowsett referring to time windows 10ka and 30ka.

| Exp. ID | $U^k_{37}$ (10ka) Foley and Dowsett (2019) | $U^k_{37}$ (30ka) Foley and Dowsett (2019) |
|---|---|---|
| Eoi400 | 3.90 | 3.72 |
| PlioM1 | 4.30 | 4.25 |
| Eoi400_K1 | 4.11 | 4.05 |

**Discussion**

1. Please also refer to the figures whenever necessary in this section.

We have revised the discussion section and have referred to figures whenever necessary.

2. Page 13, line 1 "effects of changes in boundary and initial conditions". I did not see that you had changed initial conditions, and if you have an effect from a change in initial conditions that means your simulation has not reached equilibrium, doesn't it?

All our simulations are well equilibrated and there is clearly a misunderstanding due to unprecise formulation on our side. We have accordingly rephrased the sentence from "effects of changes in boundary and initial conditions" to "effects of changes in boundary conditions".

3. Please revise the discussion regarding the effect of 5 ppm $CO_2$ change on the SST after you have found a way to remove the artifact of the cold signal in the North Atlantic, either by averaging on longer time period, or by running the model to full equilibrium regarding the NADW formation.

We thank the reviewer for suggesting possible ways to remove the supposed cold signal in the North Atlantic. We have averaged the simulation over a longer time period and the discussion regarding the effect of 5 ppm $CO_2$ has been changed.

4. Page 16 discussion on sea-ice : "may tell very different stories about the evolution of sea-ice". Certainly, different models lead to different sea-ice simulations. However, what I conclude from your results is, a small change in forcing leads to small changes in sea-ice, but the big story in the same in all you PlioMIP simulations. With COSMOS, in Pliocene conditions, you have strongly reduced sea ice with almost sea-ice free summers in PlioM1, Eoi400, Eoi405, Eoi400_ORB and Eoi405_ORB, and a remarquably similar winter sea-ice extent for all these simulations and

Eoi400_K1. Slightly more Arctic sea ice in summer with Eoi400_K1. To me, all these simulations, except maybe Eoi400_K1 which has slightly more ice, tell the same story of sea-ice. But these changes are anyway much smaller than the precision that sea-ice proxies can provide.

We agree with the reviewer that the changes are small, and this part of the discussion section on sea ice has been removed.

5. By the way what is sea-ice compactness? Did you mean sea-ice thickness? I have never seen sea-ice compactness before.

The analyzed model variable is sea ice compactness. It is the fraction of sea ice covered to sea ice free ocean surface at any grid cell. We have changed "sea ice compactness" to the more widely used term "sea ice concentration".

**Conclusion**

1. Page 17 Please update the conclusions regarding the effect of 5 ppm $CO_2$ change in the North Atlantic, according to the comments above.

Updated.

I hope that my comments are helpful to the authors. Sincerely

Yes, they have been extremely helpful in improving our manuscript. Thanks for your time and best regards.

**References**

Foley, K. M., and Dowsett, H.J.: Community sourced mid-Piacenzian sea surface temperature (SST) data: U.S. Geological Survey data release, https://doi.org/10.5066/P9YP3DTV, 2019.

McClymont, E. L., Ford, H. L., Ho, S. L., Tindall, J. C., Haywood, A. M., Alonso-Garcia, M., Bailey, I., Berke, M. A., Littler, K., Patterson, M., Petrick, B., Peterse, F., Ravelo, A. C., Risebrobakken, B., De Schepper, S., Swann, G. E. A., Thirumalai, K., Tierney, J. E., van der Weijst, C., and White, S.: Lessons from a high CO2 world: an ocean view from ~ 3 million years ago, Clim. Past Discuss., https://doi.org/10.5194/cp-2019-161, in review, 2020.

---

## Author Comment (AC2) · 29 May 2020

**Reply to comments by anonymous referee #2**

We would like to thank anonymous referee #2 for his/her time and effort in reviewing our manuscript (cp-2020-05). The comments raised in the review are highly appreciated and have helped us to further improve our manuscript. In the following, we respond to comments raised in the review.

**GENERAL COMMENTS**

1. The title seems to be not so consistence with the introduction. From the title, I think the changes in orbit forcing and PlioMIP's boundary conditions are all the main topic, even the orbit forcing (basically orbit forcing is one of the PlioMIP's boundary conditions) is more important. However, from the introduction, only the last 5 lines (line 27-31 in page 3) you mention the orbit forcing. And you only did one orbit forcing experiment, that gives me a feeling whether it's necessary to put the orbit forcing in this paper. Maybe only analyzing the impact of the changes in PlioMIP's boundary conditions to mid-Pliocene climate could be better.

The scope of this paper extends beyond comparing PlioMIP1&2 simulations, as we took the initiative to compare with another orbit which is not specified within the framework of PlioMIP. Hence, we show that a change of orbit can impact the results of PlioMIP2. We believe that the results on the alternative orbit, that provides a regionally and seasonally warmer but equally plausible climate in the context of the mid-Pliocene, are valuable additions to our contribution to PlioMIP2. Consequently, we suggest to keep these as a part of the manuscript. In particular, we see that the North Atlantic Ocean is warmer in K1 than in KM5c, and we aim at testing to which extent the model-data agreement is influenced by the choice of the orbit. Hence, we propose that the sensitivity study based on one specific warmer orbit highlights the temperature variability across the mid-Pliocene, and potentially may be a starting point towards future model-data comparisons that are based on other time-slices than KM5c. We agree that the introduction section might not convey this information concisely, so we have elaborated more on the orbital forcing in the introduction section.

2. Additionally, I think precipitation and temperature are two basic climate variable. Adding some analyse of the precipitation could be better.

The main idea going into this study was to infer the major driver of the mid-Pliocene warmth, hence the sensitivity studies by changing different boundary conditions and analyzing SAT and SST. According to suggestions of both reviewers, we have added some precipitation analyses showing the difference between PlioMIP1 and PlioMIP2 core simulations in supplement 1.

**SPECIFIC COMMENTS**

1. Page 2, line 25: delete "and the Pleistocene-Pliocene". You only study the mid-Pliocene period, never investigate anything during the Pleistocene-Pliocene period.
Pleistocene-Pliocene deleted.

2. Page 5, line 6-8: the sentence "that is useful......that is based......", there are two "that is".

We have reformulated the sentence.
From
Thus, the KM5c time-slice has been selected, partly on the basis of strong similarity of the orbit at that time to the modern orbital configuration, that is useful towards interpreting paleoenvironments in the context of future warming (Haywood et al., 2016) that is based on anthropogenic activity and will obviously be set in a near-modern orbital configuration.

To

Thus, the KM5c time-slice has been selected, partly on the basis of strong similarity of the orbit at that time to the modern orbital configuration. This is useful towards interpreting paleoenvironments in the context of future warming (Haywood et al., 2016) that is based on anthropogenic activity and will obviously be set in a near-modern orbital configuration.

3. Page 5, line 19-21: I think there are three major components, not four.
The ocean biogeochemistry model HAMOCC is an important component of the COSMOS. We agree that writing it in another sentence different from that which lists ECHAM, MPI-OM and JSBACH may confuse readers, so we have included it accordingly.

4. Consistency: the authors use "PI" instead of "pre-Industrial", but not all. I think except the first one, the others all use PI could be better. In figure 2, 3, 4, 5 and the main body, the temperature's unit is K; but in figure 6, 8, it is degree C. Maybe all use degree C could be better.
- We have changed all "pre-industrial" to "PI" except the first one.
- Temperature unit K has been changed to degree Celsius to ensure consistency.

5. Some paragraphs are really too long (e.g. the first paragraph in SAT, the third paragraph in discussion), it's really hard for readers to follow (at least for me). I suggest the authors to divide those long paragraphs into two or three shorter paragraphs.
We agree that the paragraphs are too long and might be hard for readers to follow, we have re-arranged the result section to address the effect of the relative contribution of boundary conditions as outlined in the by RC1. Long paragraph have also been shortened to make it more comprehensible.

6. Page 17, line 27-29, the sentence "The difference in CO2 between PlioMIP1 and PlioMIP2 simulations does not change the general impression of large scale mPWP climate patterns, but produces warmer oceans especially in high latitudes of the Northern Hemisphere" is not so clear. I don't know which one produces the warmer oceans, the PlioMIP1 or PlioMIP2?
Modified to enable clarity.

"The difference in $CO_2$ between PlioMIP1 and PlioMIP2 simulations does not change the general impression of large scale mPWP climate patterns, but produces warmer oceans in PlioMIP2 especially in high latitudes of the Northern Hemisphere"

7. What's the "sea ice compactness" mean? In the results, the authors use "sea ice extent and compactness", but in the discussion, "sea ice extent and compactness" and "sea ice extent and thickness" are all used. Does that mean "compactness" equal to "thickness"?

The analyzed model variable is sea ice compactness. It is the fraction of sea ice cover to sea ice free ocean surface at any grid cell. We have changed "sea ice compactness" to the more widely used "sea ice concentration".

8. As a reader, I think the discussion is not so clear and logically organized.

In the discussion section, long paragraphs have either been shortened or splitted into more comprehensible paragraphs.

**TECHNICAL CORRECTIONS**
page 9, line 17, use "Hemisphere, we find" instead of "Hemisphere we find";
Fixed

page 11, line30, use "Figure 11f, h" instead of "Figure 11f and 11h";
Fixed

page 12, line 1, use "(see Figure 12b, d)" instead of "(Figure 12b and 12d";
Fixed

page 12, line 8, use "(compare Figure 12b, d, f, h with 11f)" instead of "(compare Figure 12b,d,f,h with 11f)";
Fixed

page 18, line 24, use " Salzmann et al., 2013), further" instead of " Salzmann et al., 2013),further".
Fixed

There are a lot these kind of mistakes, please check the whole paper carefully.
We thank the reviewer for pointing these mistakes out. The whole paper has been carefully checked and all the missing spaces, commas and brackets have been fixed.

---

## Author Response (AR1)

*Author's comments*
*to*
*cp-2020-5*
**"Sensitivity of mid-Pliocene climate to changes in orbital forcing, and PlioMIP's boundary conditions"**

We want to sincerely thank both reviewers for their positive reviews and suggesting ways to improve our manuscript. All comments raised are highly valuable and has helped us to improve our manuscript, substantially.  We have implemented all suggestions into a revised manuscript. Below, we show highlights of changes made, based on the reviewer's comments.

- Anonymous reviewer #1 raised one major comment about our sensitivity tests to changing $CO_2$ from 405 to 400 ppm, and the large difference in the North Atlantic SST due to this change. For this, we have carried out additional analyses which show that both simulations are in a quasi-steady state before being analyzed (see Figure S1), and we implemented the reviewer's suggestion by  averaging over a longer time period (Figure S2).

- Other changes in the manuscript can be found in the results section, where we have re-arranged the subsections. Previously, we described the results in terms of climatic variables, namely SAT, SST and sea ice. To make the manuscript easier to follow, we describe the results in the following order;

1. Comparison of selected climatic variables between PlioMIP1 and PlioMIP2

2. Contributions of palaeogeography and changes in $CO_2$ to PlioMIP2

3. Effect of alternative orbit (MIS K1) on mid-Pliocene simulations

- Furthermore, additional analyses are carried out to compare simulated mid-Pliocene SSTs with emerging time-slice reconstructions by Foley and Dowsett (2019), and McClymont et al., 2020. The results are presented in Table 2.

We provide a list of other changes that have been implemented in the revised manuscript:

- ✔ A part of the abstract detailing minor changes in boundary conditions are deleted
- ✔ We have added some precipitation analyses showing the difference between PlioMIP1 and PlioMIP2 core simulations in the supplement (Figure S3)
- ✔ We fixed various typographical errors, missing comas and brackets
- ✔ We added 2 new references
- ✔ We added additional author's affiliation
- ✔ We replaced all occurrences of Pre-industrial by PI except the first one
- ✔ For consistency, we have replaced all occurrences of the temperature unit K by $^{\circ}$C
- ✔ Following the suggestions of both reviewers, we have replaced the term "sea ice compactness" with "sea ice concentration"
- ✔ We added additional text in the experimental design section detailing integration length and the period that we analyzed.
- ✔ We have called the simulations by their ID other than describing them at every possible occasion
- ✔ We fixed formulations that are not sufficiently precise in the discussion section, according to the reviewer's comments

✔ We added additional texts to the introduction section based on the comment by reviewer #2

**References:**

Foley, K. M., and Dowsett, H.J.: Community sourced mid-Piacenzian sea surface temperature (SST) data: U.S. Geological Survey data release, https://doi.org/10.5066/P9YP3DTV, 2019.

McClymont, E. L., Ford, H. L., Ho, S. L., Tindall, J. C., Haywood, A. M., Alonso-Garcia, M., Bailey, I., Berke, M. A., Littler, K., Patterson, M., Petrick, B., Peterse, F., Ravelo, A. C., Risebrobakken, B., De Schepper, S., Swann, G. E. A., Thirumalai, K., Tierney, J. E., van der Weijst, C., and White, S.: Lessons from a high CO2 world: an ocean view from ~ 3 million years ago, Clim. Past Discuss., https://doi.org/10.5194/cp-2019-161, in review, 2020.